# Multidimensional Adaptive Coefficient
# for Inference Trajectory Optimization in Flow and Diffusion

**Dohoon Lee** [1 2]   **Jaehyun Park** [1]   **Hyunwoo J. Kim** [3]   **Kyogu Lee** [1 2 4]

## Abstract

Flow and diffusion models have demonstrated strong performance and training stability across various tasks but lack two critical properties of simulation-based methods: freedom of dimensionality and adaptability to different inference trajectories. To address this limitation, we propose the Multidimensional Adaptive Coefficient (MAC), a plug-in module for flow and diffusion models that extends conventional unidimensional coefficients to multidimensional ones and enables inference trajectory-wise adaptation. MAC is trained via simulation-based feedback through adversarial refinement. Empirical results across diverse frameworks and datasets demonstrate that MAC enhances generative quality with high training efficiency. Consequently, our work offers a new perspective on inference trajectory optimality, encouraging future research to move beyond vector field design and to leverage training-efficient, simulation-based optimization.

## 1. Introduction

Compared to simulation-based methods like NeuralODE (Chen et al., 2018), flow and diffusion modeling (Sohl-Dickstein et al., 2015; Lipman et al., 2023) demonstrates remarkable performance and training stability across various tasks and has become a standard approach for generation tasks. However, it trades off two critical properties that simulation-based methods offer, which could enhance the quality of transportation: freedom of dimensionality and adaptability with respect to different inference trajectories. While simulation-based methodologies possess these prop-

---

[1] Music and Audio Research Group, Department of Intelligence and Information, Seoul National University, Seoul, Korea [2] Interdisciplinary Program in Artificial Intelligence, Seoul National University, Seoul, Korea [3] School of Computing, KAIST, Daejeon, Korea [4] Artificial Intelligence Institute, Seoul National University, Seoul, Korea. Correspondence to: Dohoon Lee <dohoonlee.research@gmail.com>.

*Proceedings of the $42^{nd}$ International Conference on Machine Learning*, Vancouver, Canada. PMLR 267, 2025. Copyright 2025 by the author(s).

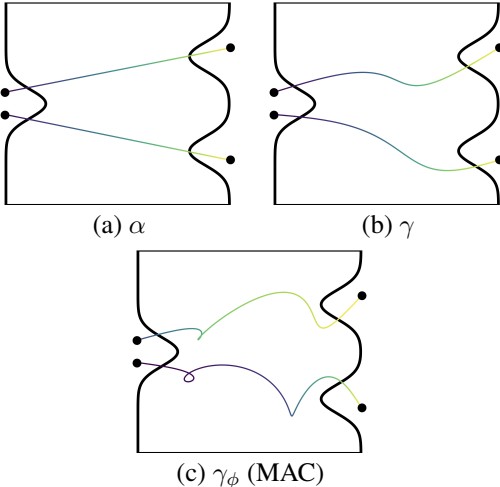

(a) $\alpha$      (b) $\gamma$

(c) $\gamma_\phi$ (MAC)

Figure 1: Comparison of $\alpha$, $\gamma$, and $\gamma_\phi$. As shown, employing $\gamma_\phi$ (MAC) expands the search space for trajectory optimization by enabling adaptive, curved trajectories and time steps that are distinct for each sample.

erties, they exhibit limitations in quality and training efficiency compared to flow and diffusion. We integrate these two properties of simulation-based methods into flow and diffusion models to enhance performance while preserving training efficiency, effectively combining their advantages.

To achieve this, we introduce the *Multidimensional Adaptive Coefficient (MAC)*. As described by Albergo et al. (2023), the trajectory with $x_0 \sim \rho_0$ and $x_1 \sim \rho_1$ in flow and diffusion for $t \in [0, T]$ can be written as $x(t) = \alpha_0(t)x_0 + \alpha_1(t)x_1$, $x_0, x_1 \in \mathbb{R}^d$, where, conventionally, the coefficients $\alpha_0(t), \alpha_1(t) \in \mathbb{R}$ are unidimensional. We extend this by introducing a *Multidimensional Coefficient* $\gamma_0(t), \gamma_1(t) \in \mathbb{R}^{d \times d}$, which allows for different time scheduling across all data dimensions. Based on this, we introduce MAC $\gamma_\phi(t, \mathbf{x}_{\theta,\phi}^{\mathcal{S}})$, parameterized by $\phi$, to adapt to different inference trajectories $\mathbf{x}_{\theta,\phi}^{\mathcal{S}}$. To optimize MAC, we use it to construct a flow- or diffusion-based differential equation solver $G_{\theta,\phi}$ based on a flow or diffusion model $H_\theta$, and adversarially optimize $\theta$ and $\phi$ using a discriminator $D_\psi$. By optimizing MAC, we achieve full multidimensionality and adaptability in the inference trajectory of flow and diffusion models. This can be applied to any type of framework as a plug-in module to enhance inference performance.

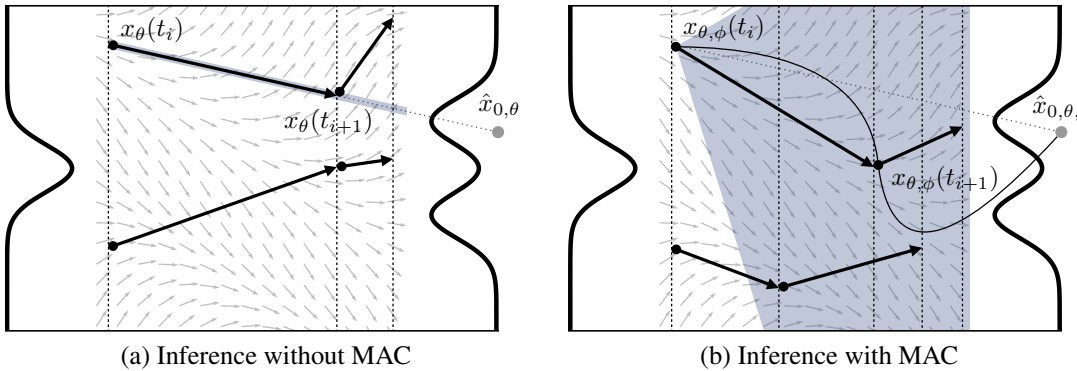

(a) Inference without MAC        (b) Inference with MAC

Figure 2: Comparison of inference trajectory optimization with and without MAC, given the same vector field estimated by $H_\theta$, indicated by gray arrows in the background. (a) Without MAC, inference trajectories strictly follow the directions defined by the vector field $H_\theta$, allowing adjustments only in step sizes, which must remain consistent across all samples. (b) With MAC, the search space (shaded in blue) expands, enabling flexible adjustments in both trajectory directions and step sizes, adaptively optimized for each individual sample. Thus, even when a suboptimal vector field is given, MAC effectively corrects errors by identifying optimal inference plans that maximize final transportation quality.

For a given vector field defined by $H_\theta$, optimizing MAC corresponds to inference trajectory optimization. In particular, optimal inference plans are computed and refined offline via simulation, and subsequently deployed during inference without incurring additional optimization costs. Before MAC, it was challenging to induce significant performance gains through inference trajectory optimization alone, due to the absence of multidimensionality and adaptability. In contrast, as shown in Figure 2, MAC substantially expands the search space for optimization by introducing dimension-wise adaptive control, thereby enhancing the effectiveness of inference trajectory optimization.

Our experiments span various frameworks (DDPM (Ho et al., 2020), FM (Lipman et al., 2023), EDM (Karras et al., 2022), SI (Albergo et al., 2023)) across multiple datasets (CIFAR-10 (Krizhevsky & Hinton, 2009), FFHQ (Karras et al., 2019), AFHQ (Choi et al., 2020), and ImageNet (Deng et al., 2009)). Notably, inference using MAC yields consistent performance improvements, achieving state-of-the-art results in CIFAR-10 conditional generation. These results are obtained with high training efficiency, suggesting that we successfully integrate the advantages of simulation-based and simulation-free dynamics.

Consequently, we propose exploring the optimality of inference trajectories through training-efficient simulation in flow- and diffusion-based generative models. Prior approaches primarily focused on shaping the vector field, often relying on predefined, simulation-free notions of optimality, such as straightness (Liu et al., 2023; Tong et al., 2024). However, the inference trajectory is governed not solely by the vector field, but by the interplay between the vector field and the inference plan. From this perspective, simulation-free vector field training deviates from true transportation optimality, which should be defined by the final quality of

transportation. Our approach addresses this limitation by introducing training-efficient simulation, thereby making simulation-based inference trajectory optimization practically feasible. In summary, our **main contributions** are:

1. We introduce the concept of a Multidimensional Adaptive Coefficient (MAC) in flow and diffusion, laying the foundation for achieving both freedom of dimensionality and adaptability in inference trajectories, easily implemented as a plug-in module for any framework.

2. We propose a highly efficient training strategy for inference trajectory optimization by leveraging MAC in combination with adversarial training.

3. We suggest exploring the notion of trajectory optimality beyond predefined properties, emphasizing evaluation based on the final transportation quality.

## 2. Related Works

**Trajectory Optimizations in Flow and Diffusion**    Various trajectory optimization approaches predefine optimality before training using simulation-free objectives. Methods such as Liu et al. (2023) and Tong et al. (2024) define straightness as the optimality criterion and optimize trajectories by maintaining consistency between $(x_0, x_1)$ when training flow and diffusion models, aligning with an Optimal Transport (OT) perspective. Other approaches, including Lee et al. (2023); Park et al. (2024); Kim et al. (2025), also adopt the OT perspective as a predefined criterion for tuning the vector field. Singhal et al. (2023) defines optimality through a fixed sequence of diffusion steps aimed at reducing inference complexity. Bartosh et al. (2024) introduces a neural flow model that implicitly sets trajectory optimality within the diffusion process. Kapuśniak et al. (2024) optimizes trajectories by defining data-dependent geodesics. There are also methods that refine trajectories after training,

such as Albergo et al. (2024), which defines optimality by minimizing trajectory length under the Wasserstein-2 metric, focusing on a shortest-distance criterion.

Despite these diverse perspectives on trajectory optimality, there are two significant differences between these methods and ours. First, we assess optimality solely based on the final transportation quality through simulation, which is a crucial factor in generative modeling. Second, none of the existing methods achieve full flexibility in inference trajectory design on two fronts: multidimensionality and adaptability with respect to different inference trajectories.

**Few-Step Generation via Distillation and Fine-Tuning**
Existing approaches can be broadly categorized into non-adversarial and adversarial methods. Non-adversarial methods, such as Geng et al. (2023); Berthelot et al. (2023); Yin et al. (2024); Song & Dhariwal (2024), focus on 1-step distillation techniques without adversarial objectives. These approaches aim for few-step generation by leveraging distributional losses and equilibrium models, effectively distilling the diffusion process without involving a discriminator. Conversely, adversarial approaches such as Wang et al. (2023); Lu et al. (2023); Luo et al. (2023); Xu et al. (2024), integrate a discriminator within diffusion models via a GAN-based framework. Kim et al. (2024) also generalizes Song et al. (2023) to enable efficient sampling using a discriminator.

While our method shares similarities in achieving few-step generation by leveraging a pre-trained diffusion model, a key difference is that prior works optimize only the vector field, omitting the inference plan. In contrast, our method jointly optimizes both the vector field and the inference plan, thereby enabling inference trajectory optimization.

# 3. Methodology

We consider the task of transporting between two distributions $x_0 \sim \rho_0$ and $x_1 \sim \rho_1$, where $x_0, x_1 \in \mathbb{R}^d$. Following Albergo et al. (2023), for $t \in [0, T]$, the trajectory $x(t)$ is defined as:

$$x(t) = \alpha_0(t)x_0 + \alpha_1(t)x_1, \quad v(t, x(t)) = \dot{x}(t), \quad (1)$$

where $\alpha(t) = [\alpha_0(t), \alpha_1(t)] \in \mathbb{R}^2$ represents the unidimensional coefficients, and $\dot{x}(t)$ denotes the derivative of $x(t)$ with respect to $t$. The diffusion model $H_\theta$ estimates the vector field $v$ as follows:

$$\begin{aligned} v_\theta(t, x(t)) &= \dot{\alpha}_0(t)\hat{x}_{0,\theta} + \dot{\alpha}_1(t)\hat{x}_{1,\theta}, \\ H_\theta(t, x(t)) &= [\hat{x}_{0,\theta}, \hat{x}_{1,\theta}]. \end{aligned} \quad (2)$$

For example, Song et al. (2021) predict the score value $\nabla_x \log p(x(t); t) = -\hat{x}_{1,\theta}/\alpha_1(t)$ to obtain the vector field. There are also flow-based methods, such as Lipman et al. (2023), that do not explicitly target $\hat{x}_{0,\theta}$ or $\hat{x}_{1,\theta}$, but instead

directly model the vector field $H_\theta(t, x(t)) = v_\theta(t, x(t))$ with the flow model $H_\theta$. All these methods achieve generative modeling by numerically solving an ODE or SDE using the predicted vector field. We use DDPM (Ho et al., 2020), FM (Lipman et al., 2023), EDM (Karras et al., 2022), and SI (Albergo et al., 2023), as summarized in Appendix A.

## 3.1. Multidimensional Adaptive Coefficient

To introduce multidimensionality into the coefficient, we first define the *multidimensional coefficient*, $\gamma(t) = [\gamma_0(t), \gamma_1(t)] \in \mathbb{R}^{d \times d \times 2}$, which generalizes the unidimensional coefficient by extending it to higher dimensions.

**Definition 3.1. (Multidimensional Coefficient)** Given a trajectory defined by $x(t) = \gamma_0(t)x_0 + \gamma_1(t)x_1$, where $x_0, x_1 \in \mathbb{R}^d$, the multidimensional coefficient is defined as:

$$\gamma(t) = [\gamma_0(t), \gamma_1(t)] : [0, T] \to \mathbb{R}^{d \times d \times 2},$$

subject to the following conditions: $\gamma_0(0) = I$, $\gamma_1(0) = \mathbf{0}_{d \times d}$, $\gamma_1(T) = TI$, and $\gamma(t) \in C^1([0, T], \mathbb{R}^{d \times d \times 2})$.

$\gamma(t) \in C^1([0, T], \mathbb{R}^{d \times d \times 2})$ indicates that $\gamma$ is continuously differentiable to the first order with respect to $t$ on the interval $[0, T]$. The boundary conditions ensure that $x(t)$ becomes $x_0$ and $x_T$ for $t = 0$ and $t = T$, respectively, which is necessary for transportation. The unidimensional coefficient $\alpha$ can be regarded as a special case of the multidimensional coefficient $\gamma$, where $\gamma$ is a scalar matrix for a given $t$.

For adaptability with respect to different inference trajectories, we define a multidimensional coefficient $\gamma_\phi$, parameterized by $\phi$, allowing it to adapt to different inference trajectories $x_{\theta,\phi}(t)$ over the inference times $\tau = \{t_0, \ldots, t_N\}$:

**Definition 3.2. (Multidimensional Adaptive Coefficient (MAC))** Let $\mathcal{S} = \{t^{(1)}, \ldots, t^{(\ell)}\} \subseteq [0, T]$ be an arbitrary set of inference times, and define the corresponding inference trajectory as $\mathbf{x}_{\theta,\phi}^{\mathcal{S}} = [x_{\theta,\phi}(t^{(1)}), \ldots, x_{\theta,\phi}(t^{(\ell)})] \in \mathbb{R}^{d \times \ell}$. Then, the MAC $\gamma_\phi(t, \mathbf{x}_{\theta,\phi}^{\mathcal{S}}) = [\gamma_{0,\phi}(t, \mathbf{x}_{\theta,\phi}^{\mathcal{S}}), \gamma_{1,\phi}(t, \mathbf{x}_{\theta,\phi}^{\mathcal{S}})]$ is defined as:

$$\gamma_\phi(t, \mathbf{x}_{\theta,\phi}^{\mathcal{S}}) : [0, T] \times \mathbb{R}^{d \times \ell} \to \mathbb{R}^{d \times d \times 2},$$

and is parameterized by $\phi$. Boundary conditions and smoothness follow Definition 3.1.

**Efficient Computation of MAC** We adopt a diagonal matrix for $\gamma_\phi$, which significantly reduces the output dimensionality and the size of the neural network for MAC, while retaining sufficient capacity to model non-linear trajectories. Accordingly, we consider $\gamma(t) \in \mathbb{R}^{d \times 2}$ instead of $\gamma(t) \in \mathbb{R}^{d \times d \times 2}$ (details in Appendix B). Furthermore, we evaluate the $\phi$-parameterized part of $\gamma_\phi$ only once at $t = T$, and reuse the resulting function throughout the entire inference schedule $\tau$. This allows us to compute $\gamma_\phi$ over the full schedule with a single forward pass through the $\phi$-network.

## 3.2. Inference Trajectory Optimization

Consider a flow- and diffusion-based differential equation solver $G_{\theta,\phi}$ with fixed configurations (e.g., NFE, integration schemes), where $\theta$ represents the vector field from the pre-trained flow or diffusion model parameters, and $\phi$ is the parameter for MAC. We propose two optimization problems. The first is:

$$\phi^* = \arg\min_{\phi} \mathbb{D}\left(\rho_0, \hat{\rho}_{0,\theta,\phi}\right), \qquad (3)$$

where $\hat{\rho}_{0,\theta,\phi}$ denotes the generated distribution from $G_{\theta,\phi}$, and $\mathbb{D}$ is a divergence metric. Solving Equation 3 corresponds to inference trajectory optimization with a fixed vector field. This setting allows us to isolate the impact of optimizing MAC and clearly demonstrate its effectiveness.

To achieve higher transportation performance, we optimize the full inference trajectory by jointly optimizing the vector field $\theta$ and the inference plan $\phi$, as follows:

$$\theta^*, \phi^* = \arg\min_{\theta,\phi} \mathbb{D}\left(\rho_0, \hat{\rho}_{0,\theta,\phi}\right). \qquad (4)$$

For low-dimensional datasets, the divergence terms above can be computed directly using the Wasserstein distance. For high-dimensional datasets, following Goodfellow et al. (2014), we minimize Equations 3 and 4 by using the following min-max problem:

$$\min_{\theta,\phi} \max_{\psi} \mathbb{E}_{x_0}\left[\log D_\psi(x_0)\right] + \mathbb{E}_{x_T}\left[\log\left(1 - D_\psi(G_{\theta,\phi}(\tau, x_T)))\right)\right],$$
$$(5)$$

where $D_\psi$ represents the discriminator, and $x_T \sim \rho_T$ is the starting point of the inference trajectory. As shown, $\theta$ and $\phi$ in $G_{\theta,\phi}$ aim to deceive $D_\psi$. Based on Equation 5, we use the simulation-based adversarial objective exclusively for $\phi$ (details in Section 3.2.3). *Optionally*, before trajectory optimization, we pre-train the flow and diffusion model $H_\theta$ using various $\gamma$ sampled from a well-designed hypothesis set $\Gamma_h$ (details in Section 3.2.2).

### 3.2.1. HYPOTHESIS SET FOR MAC

We define the multidimensional coefficient set $\Gamma$ as:

$$\Gamma = \left\{ \begin{array}{l} \gamma(t): [0,T] \to \mathbb{R}^{d\times 2}, \quad \text{where} \\ \gamma(t) = [\gamma_0(t), \gamma_1(t)], \ \ \gamma(t) \in C^1([0,T], \mathbb{R}^{d\times 2}), \\ \gamma_0(0) = \mathbf{1}_d, \ \gamma_1(0) = \mathbf{0}_d, \ \gamma_1(T) = T\mathbf{1}_d \end{array} \right\}.$$
$$(6)$$

To design the hypothesis set $\Gamma_h \subseteq \Gamma$ for exploring $\gamma_\phi$, we consider three main properties. First, the hypothesis set should be broad enough to include the optimal coefficient $\gamma_{\phi^*}$ while avoiding unnecessary complexity to reduce the burden on the flow and diffusion model. As shown in Figure 3, some multidimensional coefficients exhibit excessive

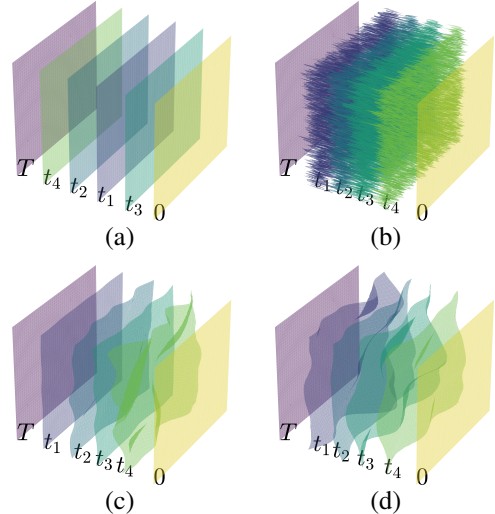

Figure 3: Crude coefficients: (a) Oscillatory behavior in $t$ due to high-frequency components; (b) High adjacent pixel differences. Refined coefficients: (c) Constrained multidimensionality for larger $t$ in pre-training; (d) Unconstrained multidimensionality for adversarial training.

high-frequency components in $t$ and across different dimensions. Given the vast size of the coefficient set, it is crucial to exclude such crude coefficients using appropriate constraints and define a well-designed hypothesis set for $\gamma_\phi$ to explore. Second, the computation of $\gamma_\phi$ via the parameter $\phi$ should require low NFE. Lastly, for pre-training the flow or diffusion models, the cost of sampling random $\gamma \sim \Gamma_h$ should be low. Considering these factors, we choose to model the weights of sinusoidals—akin to a Fourier expansion—by parameter $\phi$, as in Albergo et al. (2024). Our chosen design is:

$$\Gamma_h = \left\{ \begin{array}{l} \gamma_\phi(t, x_T): [0,T] \times \mathbb{R}^d \to \mathbb{R}^{d\times 2}, \quad \text{where} \\ \gamma_\phi(t, x_T) = [\gamma_{0,\phi}(t, x_T), \gamma_{1,\phi}(t, x_T)], \\ \gamma_{0,\phi}(t, x_T) = \mathcal{F}_0(b_m(t), w_\phi(x_T)), \\ \gamma_{1,\phi}(t, x_T) = \mathcal{F}_1(b_m(t), w_\phi(x_T)), \ \phi \in \mathcal{P} \end{array} \right\},$$
$$(7)$$

where $\mathcal{P}$ represents the parameter set for $\phi$. The function $\mathcal{F}$ constructs the weighted sinusoidal expansion using basis functions $b_m(t)$ and their weights $w_\phi$, defined as:

$$b_m(t) = \sin\left(\pi m(t/T)^{1/q}\right) \in \mathbb{R}, \quad q \in \mathbb{R}, \qquad (8)$$
$$w_\phi(x_T) = s\, \text{LPF} \circ \tanh\left(\text{NN}_\phi(x_T)\right), \quad s \in \mathbb{R}, \qquad (9)$$

where $w_\phi(x_T) \in \mathbb{R}^{d\times M\times 2}$ represents the multidimensional weights for the sinusoidals, modeled by the neural network $\text{NN}_\phi$. LPF refers to low-pass filtering, implemented as convolution with a Gaussian kernel, applied across different $d$ dimensions to exclude high frequencies. This parameterization always satisfies $\Gamma_h \subseteq \Gamma$. For image generation, we employ a U-Net $U_\phi$ for $\text{NN}_\phi$, and condition it as $U_\phi(x_T, c)$ for conditional generation. Details are provided in Appendix C.

This parameterization satisfies all three properties mentioned above. First, high-frequency components in $t$ and $d$ can be excluded by controlling hyperparameters such as $s$ and LPF configurations. Second, $\gamma_\phi$ can be computed with 1 NFE using $\mathrm{NN}_\phi$. Lastly, because the $\tanh$ in Equation 9 bounds the output of $\mathrm{NN}_\phi$ to $(-1, 1)$, we can efficiently sample a random multidimensional coefficient $\gamma(t, u) \sim \Gamma_h$ by directly sampling sinusoidal weights $w$ from a uniform distribution rather than evaluating $\mathrm{NN}_\phi$ during pre-training. Specifically, $w(u) = s\,\mathrm{LPF} \circ u,\ u \sim \mathcal{U}(-1, 1) \in \mathbb{R}^{d \times M \times 2}$.

### 3.2.2. Optional $\gamma$-Pre-Training for $H_\theta$

To ensure that $H_\theta$ can better accommodate $\gamma$, we introduce a pre-training procedure. The loss functions for pre-training $H_\theta$ follow the original flow and diffusion losses, except that $\gamma$ is used instead of $\alpha$, and coefficient conditioning is incorporated. For example, in the case of DDPM, the loss function is given by $\mathcal{L}_\theta^{\mathrm{pre}} = \mathbb{E}_{t,x_0,x_1,u}\|H_\theta(t, x(t), \gamma(t, u)) - x_0\|_2^2$. Detailed versions of the loss functions for each flow and diffusion framework are provided in Appendix E.

**Coefficient Conditioning**  When adopting existing pre-trained models for $H_\theta$, we apply $H_\theta(\bar{\gamma}_\phi(t, x_T), x(t))$ during trajectory optimization, where $\bar{\gamma}_\phi(t, x_T) \in \mathbb{R}$ denotes the scalar average of $\gamma_\phi(t, x_T)$ over all dimensions. However, to allow $\gamma_\phi$ to receive precise gradients through $H_\theta$, we incorporate $\gamma_\phi(t, x_T)$ directly into the model during inference trajectory optimization by using $H_\theta(t, x(t), \gamma_\phi(t, x_T))$. To support this without modifying the structure of $H_\theta$, we concatenate $\gamma_\phi(t, x_T)$ with $x(t)$ along the channel axis as input to $H_\theta$. We also pre-train $H_\theta$ with coefficient conditioning in the form $H_\theta(t, x(t), \gamma(t, u))$, consistent with its usage during inference trajectory optimization.

**$\Gamma_h$ for Pre-Training**  Given that a large $\Gamma_h$ of coefficients for pre-training can burden $H_\theta$ and potentially degrade performance, we use a smaller $\Gamma_h$ during pre-training and fully enable multidimensionality across $t$ during adversarial training, as shown in Figure 3 (c) and (d). Specifically, we use $\gamma$ with high multidimensionality near $t = 0$ and low multidimensionality at larger $t$ by configuring the LPF during the pre-training of $H_\theta$. For adversarial training, we fully enable multidimensionality for $\gamma_\phi$ across the entire range of $t$. Details are provided in Appendix C.

**Optionality of Pre-Training**  We emphasize that this pre-training stage is *optional*, as MAC can effectively leverage existing models pre-trained with $\alpha$ without substantially compromising performance. While using $\gamma$ for pre-training increases the probability of generating multidimensional interpolated values $x(t)$, multidimensional interpolation inherently arises even with a standard coefficient $\alpha$, since $x_1 \sim \rho_1 = \mathcal{N}(0, I) \in \mathbb{R}^d$ implies that each dimension $x_{1,i}$ is independently drawn from $\mathcal{N}(0, 1)$. Consequently, for large-scale datasets where training from scratch is computa-

tionally prohibitive, MAC remains compatible with existing pre-trained models, ensuring practical applicability.

### 3.2.3. Adversarial Optimization

For Equation 5, we use the hinge loss (Lim & Ye, 2017) with the StyleGAN-XL (Sauer et al., 2022) discriminator for $D_\psi$, as used in Kim et al. (2024). Details for the flow- and diffusion-based differential equation solver $G_{\theta,\phi}$ are provided in Appendix D. We use separate loss functions for $\theta$ and $\phi$, applying the simulation-based objective exclusively to $\phi$ as follows:

$$
\begin{aligned}
\mathcal{L}_\phi &= -\mathbb{E}_{x_T}[D_\psi(G_{\theta,\phi}(\tau, x_T))], \\
\mathcal{L}_\psi &= \mathbb{E}_{x_0}[\max(0, 1 - D_\psi(x_0))] \\
&\quad + \mathbb{E}_{x_T}[\max(0, 1 + D_\psi(G_{\theta,\phi}(\tau, x_T)))],
\end{aligned}
\tag{10}
$$

where $\mathcal{L}_\phi$ and $\mathcal{L}_\psi$ indicate that gradients are computed with respect to $\phi$ and $\psi$, respectively. Through these loss functions, $\phi$ is optimized to determine better coefficients via simulation through $G_{\theta,\phi}$. For $\theta$, we use the adversarial loss defined as:

$$
\begin{aligned}
\mathcal{L}_\theta &= -\mathbb{E}_{t,x_0,x_1,z}\left[D_\psi\left(H_\theta(t, x(t), \gamma_\phi(t, z))\right)\right], \\
x(t) &= \gamma_{0,\phi}(t, z) \odot x_0 + \gamma_{1,\phi}(t, z) \odot x_1,
\end{aligned}
\tag{11}
$$

where $z \sim \rho_T$ (sampled like $x_T$, used only to train $H_\theta$), $x_0 \sim \rho_0$, $x_1 \sim \rho_1$, $t \sim \tau$, and $H_\theta(\bar{\gamma}_\phi(t, z), x(t))$ is used when coefficient conditioning is disabled. Given that $H_\theta$ approximates $x_0$, it can be adversarially optimized with $D_\psi$. Since $H_\theta$ only needs to handle coefficients from $\gamma_\phi$ rather than the full hypothesis set $\Gamma_h$, it is trained exclusively using $\gamma_\phi$, thereby reducing its computational burden, as $\gamma_\phi$ is significantly smaller than $\Gamma_h$. The final loss terms are:

$$
\mathcal{L}_{\phi,\psi}^{\mathrm{adv}} = \mathcal{L}_\phi + \mathcal{L}_\psi,
\tag{12}
$$

$$
\mathcal{L}_{\theta,\phi,\psi}^{\mathrm{adv}} = \mathcal{L}_\theta + \mathcal{L}_\phi + \mathcal{L}_\psi.
\tag{13}
$$

Here, Equation 12 corresponds to Equation 3, and Equation 13 corresponds to Equation 4. The procedure corresponding to Equation 13 is summarized in Algorithm 1.

---

**Algorithm 1** $\gamma$-Pre-Training and Adversarial Optimization

---

**Input:** $\rho_0, \rho_1, \rho_T, H_\theta, \gamma_\phi, G_{\theta,\phi}, D_\psi, \mathcal{T}, \tau$
**if** $\gamma$-pretraining **then**
  **repeat**
    Sample $x_0 \sim \rho_0,\ x_1 \sim \rho_1,\ u \sim \mathcal{U}(-1, 1),\ t \sim \mathcal{T}$
    $[\hat{x}_{0,\theta}, \hat{x}_{1,\theta}] \leftarrow H_\theta(t, x(t), \gamma(t, u))$
    Compute $\mathcal{L}_\theta^{\mathrm{pre}}$; Update $\theta$
  **until** $\theta$ converges
**end if**
**repeat**
  Sample $x_0 \sim \rho_0,\ x_1 \sim \rho_1,\ x_T \sim \rho_T,\ z \sim \rho_T,\ t \sim \tau$
  $\hat{x}_{0,\theta,\phi} \leftarrow \begin{cases} H_\theta(t, x(t), \gamma_\phi(t, z)) & \text{with } \gamma\text{-pretraining} \\ H_\theta(\bar{\gamma}_\phi(t, z), x(t)) & \text{w/o } \gamma\text{-pretraining} \end{cases}$
  $x_{\theta,\phi}(t_N) \leftarrow G_{\theta,\phi}(\tau, x_T)$
  Compute $\mathcal{L}_{\theta,\phi,\psi}^{\mathrm{adv}}$; Update $\theta, \phi, \psi$
**until** $\theta,\ \phi,\ \psi$ converge

---

Table 1: $\mathcal{W}_2$ distance ($\downarrow$) for 2-dimensional transportation results. The best performance is highlighted.

| Method \ NFE | Gaussian to 8 Gaussians | | Gaussian to Moons | | 8 Gaussians to Moons | | Moons to 8 Gaussians | |
|---|---|---|---|---|---|---|---|---|
| | 5 | 10 | 5 | 10 | 5 | 10 | 5 | 10 |
| $SI_\alpha$ | $0.763_{\pm0.040}$ | $0.673_{\pm0.055}$ | $0.882_{\pm0.035}$ | $0.643_{\pm0.060}$ | $0.981_{\pm0.112}$ | $0.649_{\pm0.165}$ | $1.271_{\pm0.185}$ | $0.998_{\pm0.203}$ |
| $SI_\gamma$ + opt $\phi$ | $0.721_{\pm0.082}$ | $0.452_{\pm0.033}$ | $0.682_{\pm0.093}$ | $0.359_{\pm0.098}$ | $0.924_{\pm0.235}$ | $0.311_{\pm0.051}$ | $0.908_{\pm0.109}$ | $0.500_{\pm0.072}$ |
| $SI_\alpha^{OT}$ | $0.457_{\pm0.021}$ | $0.440_{\pm0.052}$ | $0.245_{\pm0.023}$ | $0.217_{\pm0.019}$ | $0.321_{\pm0.064}$ | $0.318_{\pm0.068}$ | $0.488_{\pm0.050}$ | $0.492_{\pm0.056}$ |
| $SI_\gamma^{OT}$ + opt $\phi$ | $0.399_{\pm0.017}$ | $0.415_{\pm0.016}$ | $0.230_{\pm0.015}$ | $0.188_{\pm0.006}$ | $0.258_{\pm0.015}$ | $0.221_{\pm0.014}$ | $0.421_{\pm0.012}$ | $0.407_{\pm0.031}$ |

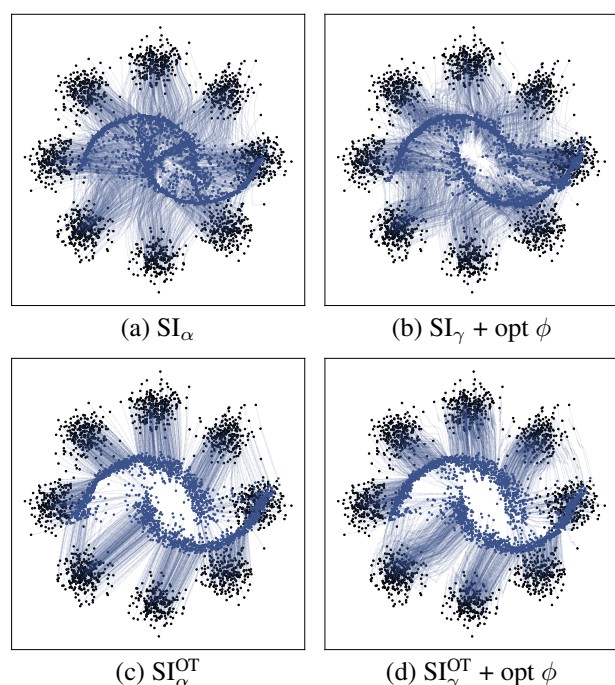

(a) $SI_\alpha$     (b) $SI_\gamma$ + opt $\phi$

(c) $SI_\alpha^{OT}$     (d) $SI_\gamma^{OT}$ + opt $\phi$

Figure 4: Comparison of inference trajectories from 8 Gaussians to Moons.

## 4. Experiments

### 4.1. 2-Dimensional Transportation: Optimizing Only $\phi$

We conduct experiments on 2-dimensional synthetic datasets used by Tong et al. (2024) with SI (Albergo et al., 2023). To validate the benefits of using MAC $\gamma_\phi$, we train $\phi$ exclusively while freezing $\theta$, as in Equation 3, ensuring a fair comparison with baseline methods. We use the Wasserstein loss function $\mathcal{L}_\phi = \mathcal{W}_2(x_0, G_{\theta,\phi}(\tau, x_T))$ instead of a discriminator. To further evaluate whether MAC improves transportation even when optimality is defined as a straight trajectory, we test additional configurations of minibatch pairing for pre-training $(x_0, x_1)$: random pairing and OT pairing (Tong et al., 2024). The minibatch-OT method encourages the flow and diffusion

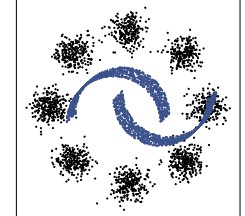

Figure 5: $x_T \sim \rho_T$ (black) and $x_0 \sim \rho_0$ (blue).

Table 2: FID ($\downarrow$) of models pre-trained with $\alpha$ and $\gamma$.

| Method \ NFE | CIFAR-10 | | | ImageNet-32 | | |
|---|---|---|---|---|---|---|
| | 100 | 150 | 200 | 100 | 150 | 200 |
| $SI_\alpha$ | 4.75 | 4.51 | 4.30 | 8.08 | 7.79 | 7.63 |
| $SI_\gamma$ | 3.98 | 3.74 | 3.63 | 6.33 | 6.21 | 6.20 |
| $FM_\alpha$ | 4.52 | 4.23 | 4.07 | 7.78 | 7.53 | 7.38 |
| $FM_\gamma$ | 3.59 | 3.42 | 3.42 | 6.18 | 6.03 | 6.01 |
| $DDPM_\alpha$ | 6.64 | 4.84 | 4.10 | 8.13 | 7.40 | 7.14 |
| $DDPM_\gamma$ | 4.73 | 4.11 | 3.83 | 6.84 | 6.51 | 6.42 |

Table 3: FID after adversarial optimization for varying NFE on CIFAR-10.

| Method \ NFE | 4 (+) | 6 (+) | 8 (+) | 10 (+) |
|---|---|---|---|---|
| $SI_\gamma$ + adv $\phi$ | 20.59 | 6.62 | 4.85 | 4.14 |
| $FM_\gamma$ + adv $\phi$ | 16.42 | 8.17 | 6.56 | 6.13 |
| $DDPM_\gamma$ + adv $\phi$ | 72.64 | 20.13 | 13.72 | 10.04 |

model to learn a straight trajectory by pairing $x_0$ and $x_1$ as an OT solution within a minibatch during pre-training, where optimality for the inference trajectory is defined as straight. Details are provided in Appendix E.1.

As shown in Table 1, using MAC consistently achieves the best results, even for models trained with minibatch-OT. This suggests that a straight trajectory is not always optimal, even in OT-trained models, and that MAC can adaptively discover better inference trajectories to correct errors. Figure 4 further illustrates how MAC adjusts the trajectory direction to optimize transportation, resulting in a path that is not straight. A comparison of (c) with (d) reveals a distinct piecewise linear trajectory in (d), indicating that a non-straight trajectory achieves superior performance. These results empirically suggest that the optimality of the inference trajectory should be defined and explored in terms of transportation quality, rather than by a predefined property.

Table 4: FID after adversarial optimization with different conditionings of $\gamma_\phi(t, \cdot)$ on CIFAR-10 (10 (+) NFE).

| Method \ Conditioning | $\mathbf{1}_d$ | $z \sim \rho_T$ | $x_T \sim \rho_T$ |
|---|---|---|---|
| $SI_\gamma$ + adv $\phi$ | 7.84 | 6.48 | 4.14 |
| $FM_\gamma$ + adv $\phi$ | 9.20 | 9.06 | 6.13 |
| $DDPM_\gamma$ + adv $\phi$ | 26.09 | 23.31 | 10.04 |

Table 5: Performance comparisons on CIFAR-10, FFHQ, AFHQ, and ImageNet. $*$ indicates metrics computed by us.

| Model | NFE | CIFAR-10 uncond. FID ($\downarrow$) | CIFAR-10 cond. FID ($\downarrow$) |
|---|---|---|---|
| **GAN** | | | |
| StyleGAN-Ada (Karras et al., 2020) | 1 | 2.92 | 2.42 |
| StyleGAN-XL (Sauer et al., 2022) | 1 | – | 1.85 |
| **Diffusion Model** | | | |
| DDPM$_\alpha$ (Ho et al., 2020) | 1000 | 3.17 | – |
| Score SDE$_\alpha$ (Song et al., 2021) | 2000 | 2.38 | 2.20 |
| EDM$_\alpha$ (Karras et al., 2022) | 35 | 1.98 | 1.79 |
| **Rectified Flow (Distillation)** | | | |
| 2-Rectified Flow$_\alpha$ (Liu et al., 2023) | 1 | 4.85 | – |
| 2-Rectified Flow$_\alpha$++ (Lee et al., 2024) | 1 | 3.07 | – |
| | 2 | 2.40 | – |
| **Consistency Model (Distillation)** | | | |
| CD$_\alpha$ (Song et al., 2023) | 1 | 3.55 | – |
| | 2 | 2.93 | – |
| CD$_\alpha$ + adv $\theta$ (Lu et al., 2023) | 1 | 2.65 | – |
| CTM$_\alpha$ (Kim et al., 2024) | 1 | 5.19 | – |
| CTM$_\alpha$ + adv $\theta$ (Kim et al., 2024) | 1 | 1.98 | 1.73 |
| | 2 | 1.87 | 1.63 |
| | 5 | 1.86$^*$ | 1.98$^*$ |
| | 6 | 1.93$^*$ | 2.04$^*$ |
| **Inference Trajectory Optimization** | | | |
| EDM$_\gamma$ + adv $\theta$, $\phi$ (Ours) | 5 (+) | 1.69$^*$ | 1.37$^*$ |

| Model | NFE | ImageNet-64 cond. FID ($\downarrow$) | FD$_{DINOv2}$ ($\downarrow$) |
|---|---|---|---|
| **GAN** | | | |
| StyleGAN-XL (Sauer et al., 2022) | 1 | 2.09 | – |
| **Diffusion Model** | | | |
| ADM$_\alpha$ (Dhariwal & Nichol, 2021) | 250 | 2.07 | – |
| EDM$_\alpha$ (Karras et al., 2022) | 79 | 2.44 | – |
| **Consistency Model (Distillation)** | | | |
| CD$_\alpha$ (Song et al., 2023) | 1 | 6.20 | – |
| | 2 | 4.70 | – |
| CTM$_\alpha$ + adv $\theta$ (Kim et al., 2024) | 1 | 1.92 | 160.8$^*$ |
| | 2 | 1.73 | 157.7$^*$ |
| | 5 | 3.02$^*$ | 195.0$^*$ |
| | 6 | 3.17$^*$ | 205.1$^*$ |
| **Inference Trajectory Optimization** | | | |
| EDM$_\alpha$ + adv $\theta$, $\phi$ (Ours) | 5 (+) | 1.48$^*$ | 70.2$^*$ |

| Model | NFE | FFHQ-64 FID ($\downarrow$) | AFHQ-64 FID ($\downarrow$) |
|---|---|---|---|
| **Diffusion Model** | | | |
| EDM$_\alpha$ (Karras et al., 2022) | 79 | 2.39 | 1.96 |
| **Rectified Flow (Distillation)** | | | |
| 2-Rectified Flow$_\alpha$++ (Lee et al., 2024) | 1 | 5.21 | 4.11 |
| | 2 | 4.26 | 3.12 |
| **Inference Trajectory Optimization** | | | |
| EDM$_\gamma$ + adv $\theta$, $\phi$ (Ours) | 5 (+) | 2.27$^*$ | 2.04$^*$ |

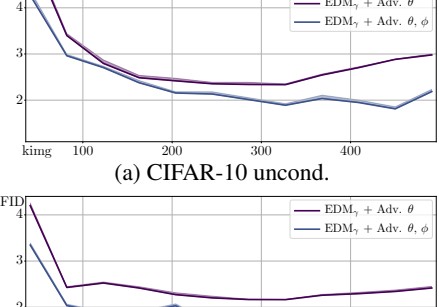

(a) CIFAR-10 uncond.

(b) CIFAR-10 cond.

Figure 6: EDM$_\gamma$ + adv $\theta$ and $\theta$, $\phi$.

Table 6: FID from the ablation study on CIFAR-10. Experiments used 500k images, compared to 1500k in Table 5.

| Configuration \ NFE | Unconditional | | Conditional | |
|---|---|---|---|---|
| | 5 (Euler) | 35 (Heun) | 5 (Euler) | 35 (Heun) |
| EDM$_\alpha$ | 68.73 | 1.97 | 48.76 | 1.79 |
| EDM$_\gamma$ | 69.58 | 2.08 | 48.53 | 1.81 |
| EDM$_\gamma$ + adv $\phi$ (no multi.) | 33.55 | – | 25.56 | – |
| EDM$_\gamma$ + adv $\phi$ | 18.67 | – | 7.77 | – |
| EDM$_\gamma$ + adv $\theta$ | 2.28 | – | 2.14 | – |
| EDM$_\gamma$ + adv $\theta$, $\phi$ | 1.81 | – | 1.42 | – |

## 4.2. Image Generation: Optimizing Only $\phi$

To isolate the benefits of using MAC in image generation, we conduct experiments that optimize only $\phi$. We employ DDPM (Ho et al., 2020), FM (Lipman et al., 2023), and SI (Albergo et al., 2023) on the CIFAR-10 (Krizhevsky & Hinton, 2009) and ImageNet-32 (Deng et al., 2009) datasets. Detailed training configurations and additional results are presented in Appendix E.2.

**Pre-Training** Before adversarial optimization, we compare the Fréchet Inception Distance (FID) (Heusel et al., 2017) of models trained with $\alpha$ and $\gamma$. Interestingly, as shown in Table 2, models trained with $\gamma$ achieve the lowest FID across all frameworks and NFE compared to models trained with $\alpha$. These results suggest that training flow and diffusion models with $\gamma \sim \Gamma_h$ and coefficient labeling can improve the innate performance of the model, despite the increased complexity compared to $\alpha$.

**Adversarial Training** We achieve approximately $10\times$ NFE efficiency across all frameworks. Specifically, in SI, we obtain FID scores of 4.14 and 7.06 on CIFAR-10 and ImageNet-32, respectively, using 10 (+) NFE (where (+) accounts for the computation of $\gamma_\phi$, given that the size of $U_\phi$ is smaller than that of $H_\theta$). These results demonstrate the compatibility of MAC across various frameworks. As shown in Table 3, MAC can also be optimized for different sampling configurations. Additionally, we validate the impact of using the starting point $x_T$ as a condition for MAC. As shown in Table 4, incorporating $x_T$ as an input to $\gamma_\phi$ consistently improves performance across all frameworks, providing empirical evidence of MAC's adaptability to $x_T$.

## 4.3. Image Generation: Optimizing $\theta$ and $\phi$

We optimize $\theta$ and $\phi$ using Equation 13 on the CIFAR-10 (Krizhevsky & Hinton, 2009), FFHQ-64 (Karras et al., 2019), AFHQv2-64 (Choi et al., 2020), and ImageNet-64

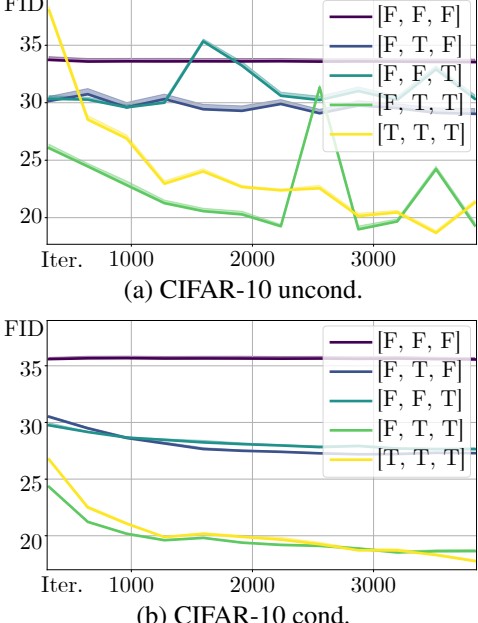

(a) CIFAR-10 uncond.

(b) CIFAR-10 cond.

Figure 7: Training with different multidimensionalities. T/F indicate whether multidimensionality is applied along the channel, height, and width axes, respectively.

(Deng et al., 2009) datasets with EDM (Karras et al., 2022). As mentioned in Section 3.2.2, since pre-training $H_\theta$ with $\gamma$ is optional for practicality, we use the existing pre-trained model $EDM_\alpha$ for ImageNet. We measure FID and $FD_{DINOv2}$ (Oquab et al., 2024). Details are provided in Appendix E.3.

**Impact of Trajectory Optimization**  As shown in Table 5, our approach generates high-quality samples across various datasets with only 5 (+) NFE, achieving a state-of-the-art result (FID = 1.37) on CIFAR-10 conditional generation. For a fair comparison in terms of NFE, we select CTM (Kim et al., 2024) due to its popularity, strong performance, and use of the same model architecture based on EDM and adversarial training with StyleGAN-XL discriminator. We then compute FID using 5 and 6 NFE on CIFAR-10 and ImageNet-64. As shown in Table 5, increasing the NFE of CTM does not significantly reduce the FID, and in some cases, the FID even increases. This indicates that MAC provides additional performance gains that cannot be achieved by distillation or vector field tuning methods alone, even with increased NFE.

To empirically validate the benefit of MAC, we conduct ablation studies by training either $\theta$ or $\phi$ individually. As presented in Table 6, the results reveal that jointly training $\theta$ and $\phi$ yields the best performance. Figure 6 further illustrates that FID decreases more rapidly during joint training compared to training $\theta$ alone. These findings suggest that performance improvements stem not only from the adversarial training of $H_\theta$, but also from the combined training of both $H_\theta$ and $\gamma_\phi$, demonstrating the benefits of MAC.

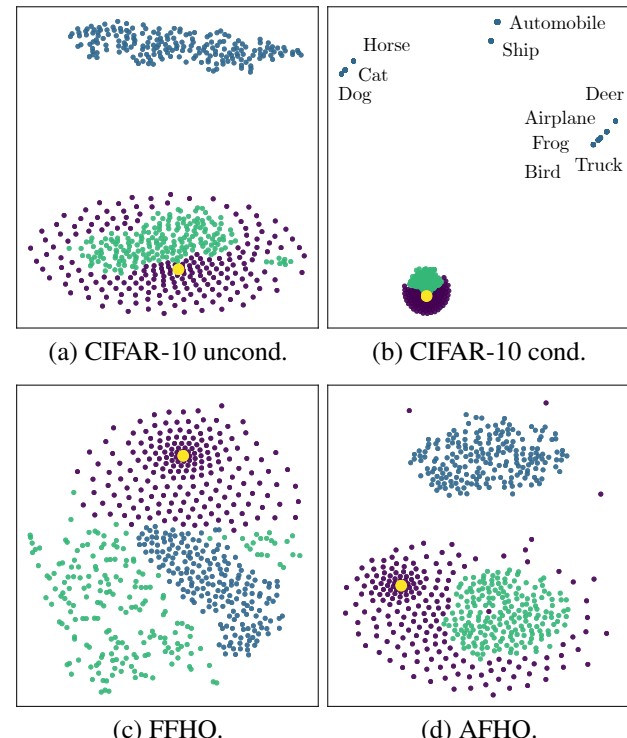

(a) CIFAR-10 uncond.

(b) CIFAR-10 cond.

(c) FFHQ.

(d) AFHQ.

Figure 8: T-SNE visualization of various coefficients. Yellow: $w = 0$ (corresponding to $\alpha$); Purple: $w = su$; Green: $w = s\,\mathrm{LPF} \circ u$ (used for pre-training); Blue: $w_\phi = s\,\mathrm{LPF} \circ \tanh(U_\phi(x_T, c))$ (trained).

**Impact of Multidimensionality of MAC**  To examine how the multidimensionality of MAC influences performance, we train $\phi$ with different levels of multidimensionality by averaging $\tanh(U_\phi(x_T))$ in Equation 9 across specific axes. For instance, to retain multidimensionality solely in the height axis ([F, T, F]), we use the same $w_\phi$ across the channel and width axes by averaging along those dimensions. As shown in Figure 7, incorporating more axes consistently improves performance, indicating that MAC's multidimensionality positively impacts generation quality.

**Analysis of Trained MAC**  To analyze how MAC is trained, we plot t-SNE embeddings of four different sinusoidal weights $w$ across multiple datasets, as shown in Figure 8. Notably, the trained $w_\phi$ diverges from weights randomly sampled from the predefined hypothesis set and is far from $\alpha$. This suggests that during adversarial training, $\gamma_\phi$ adaptively identifies optimal coefficients without heavily depending on the pre-trained distribution of $\gamma \sim \Gamma_h$. Interestingly, $\gamma_\phi$ exhibits a sparser distribution in CIFAR-10 conditional generation than in the unconditional setting, with nearly identical $w_\phi$ values for samples sharing the same label condition $c$. This indicates that the optimality of the coefficient depends more on the label condition $c$ than on the starting point $x_T$ of the differential equation.

**Training Efficiency** Our method demonstrates notable training efficiency despite incorporating simulation-based training. As shown in Table 7, the number of training images required by our approach is significantly lower across all datasets—approximately 20 to 2000 times fewer than distillation-based methods such as CTM.

Training times are 10, 2, 6, and 9 hours for CIFAR-10, FFHQ, AFHQ, and ImageNet, respectively, which are also comparatively low. The primary cost of simulation dynamics arises from VRAM requirements. However, this remains practically feasible, as strong performance can be achieved with just 5 (+) NFE. These results not only highlight the effectiveness of simulation-based optimality, but also showcase the strength of combining simulation-free and simulation-based methodologies—leveraging the advantages of each while mitigating their limitations.

Table 7: kimg ($\downarrow$) for adversarial training.

| Dataset | $\text{CTM}_\alpha$ + adv $\theta$ | Ours |
|---------|------|------|
| CIFAR-10 | 25.6k | 1.5k |
| FFHQ | 38.4k | 1.0k |
| AFHQ | 51.2k | 1.0k |
| ImageNet | 61.4k | 30 |

## 5. Conclusions

We have introduced the Multidimensional Adaptive Coefficient (MAC), a plug-in module for flow- and diffusion-based generative models that enables both freedom of dimensionality and adaptability to different inference trajectories—properties previously found only in simulation-based generative modeling. Our method improves generative performance across various datasets and frameworks with high training efficiency. These results highlight the potential of inference trajectory optimization with MAC via simulation—a direction that has been underexplored compared to vector field design. We suggest broadening the notion of trajectory optimality beyond predefined criteria such as straightness, toward a more general view based on final transportation quality. We hope our work encourages further exploration.

## 6. Future Directions

First, the current implementation of $\gamma_\phi$ is tied to a fixed NFE during trajectory optimization. Future work could explore adaptive control of NFE based on inference trajectories. Second, while $\gamma_\phi$ currently conditions only on $x_T$, future extensions could, with appropriate design, condition on trajectories $\mathbf{x}_{\theta,\phi}^S$ over diverse inference times and incorporate other contextual signals. Third, optimizing MAC under alternative objectives beyond distributional divergence may broaden its applicability to domains where task-specific criteria are more relevant. Finally, interpreting MAC selection as a form of trajectory-level policy learning may inspire novel approaches, particularly when combined with ideas from control theory or reinforcement learning.

## Impact Statement

Our research introduces the Multidimensional Adaptive Coefficient (MAC), enhancing the generative quality and efficiency of flow and diffusion models. The broader implications of this advancement are twofold. On the positive side, MAC improves the accessibility and practicality of generative models, potentially benefiting diverse applications such as medical imaging, drug discovery, and creative industries by reducing computational costs and enhancing output quality. However, improved generative technologies also carry potential risks, including the misuse in creating deepfakes or synthetic biological threats. It is essential for future work to develop robust safeguards and detection methods to mitigate such harmful applications. Continued interdisciplinary efforts in governance, ethics, and technical controls will be critical to ensuring the beneficial integration of our contributions into society.

## Acknowledgement

This work was supported by Institute of Information & communications Technology Planning & Evaluation (IITP) grant funded by the Korea government(MSIT) [NO.RS-2021-II211343, Artificial Intelligence Graduate School Program (Seoul National University)]

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

## Outline of Appendix

Appendix A provides an overview of flow and diffusion frameworks. Appendix B provides justification for the use of diagonal matrices in the design of $\gamma_\phi$. Appendix C describes the hypothesis set for MAC. Appendix D presents details of the differential equation solver $G_{\theta,\phi}$. Appendix E outlines experimental setups, including datasets, training configurations, and additional results. Appendix F explains evaluation metrics, and Appendix G provides visualizations of generated samples and inference trajectories.

## A. Flow and Diffusion Frameworks

During inference, all frameworks transport samples between $\rho_T$ and $\rho_0$, either in the forward or reverse direction.

**Denoising Diffusion Probabilistic Models (DDPM)**   DDPM (Ho et al., 2020) is one of the most widely used diffusion-based generative frameworks. Given $x_0 \sim \rho_0 = \mathcal{N}(0, I)$ and $x_1 \sim \rho_1$ as data, the VP diffusion coefficient is:

$$\alpha(t) = [\alpha_0(t), \alpha_1(t)] = \left[\sqrt{1 - b_{1-t}^2}, b_{1-t}\right], \quad b_t = e^{-\frac{1}{2}\int_0^t c(s)ds}, \quad c(s) = c_{\min} + s(c_{\max} - c_{\min}), \quad T = 1, \quad (14)$$

where $c_{\min} = 0.1$ and $c_{\max} = 20$. DDPM minimizes the loss $\mathcal{L}_\theta = \mathbb{E}_{t,x_0,x_1}\left[\|H_\theta(t, x(t)) - x_0\|_2^2\right]$, where $t \sim \mathcal{U}(0, 1)$.

**Flow Matching (FM)**   FM (Lipman et al., 2023) introduces a simple yet effective objective for generation. Given $x_0 \sim \mathcal{N}(0, I)$ and $x_1 \sim \rho_1$ as data, it employs the following coefficient:

$$\alpha(t) = [\alpha_0(t), \alpha_1(t)] = [1 - (1 - \sigma_{\min})t, t], \quad T = 1. \quad (15)$$

FM directly models $v$ using the objective $\mathcal{L}_\theta = \mathbb{E}_{t,x_0,x_1}\left[\|H_\theta(t, x(t)) - (x_1 - (1 - \sigma_{\min})x_0)\|_2^2\right]$, where $t \sim \mathcal{U}(0, 1)$.

**Elucidating Diffusion Model (EDM)**   EDM (Karras et al., 2022) refines and stabilizes diffusion model training. Given $x_0 \sim \rho_0$ as data and $x_1 \sim \rho_1 = \mathcal{N}(0, I)$, the coefficient is defined as:

$$\alpha(t) = [\alpha_0(t), \alpha_1(t)] = [1, t], \quad T = 80. \quad (16)$$

The loss function is:

$$\mathcal{L}_\theta = \mathbb{E}_{t,x_0,x_1}\left[\lambda(t)c_{\text{out}}(t)^2 \left\| F_\theta\left(c_{\text{noise}}(t), c_{\text{in}}(t)x(t)\right) - \frac{1}{c_{\text{out}}(t)}(x_0 - c_{\text{skip}}(t)x(t)) \right\|_2^2\right], \quad (17)$$

where:

$$c_{\text{in}}(t) = \frac{1}{\sqrt{\alpha_1^2(t) + \sigma_{\text{data}}^2}}, \quad c_{\text{out}}(t) = \frac{\alpha_1(t) \cdot \sigma_{\text{data}}}{\sqrt{\sigma_{\text{data}}^2 + \alpha_1^2(t)}}, \quad c_{\text{skip}}(t) = \frac{\sigma_{\text{data}}^2}{\alpha_1^2(t) + \sigma_{\text{data}}^2},$$
$$c_{\text{noise}}(t) = \frac{1}{4}\ln t, \quad \lambda(t) = \frac{\alpha_1^2(t) + \sigma_{\text{data}}^2}{(\alpha_1(t) \cdot \sigma_{\text{data}})^2}, \quad (18)$$

with $t$ sampled from $\ln(t) \sim \mathcal{N}(-1.2, 1.2^2)$, and $\sigma_{\text{data}} = 0.5$. EDM models $H_\theta(t, x(t)) = c_{\text{skip}}(t)x(t) + c_{\text{out}}(t)F_\theta(c_{\text{noise}}(t), c_{\text{in}}(t)x(t)) = \hat{x}_{0,\theta}$. During inference, EDM transports from $\rho_T = \mathcal{N}(0, T^2I)$ to $\rho_0$.

**Stochastic Interpolant (SI)**   SI (Albergo et al., 2023) facilitates transportation between arbitrary distributions $\rho_0$ and $\rho_1$. The conventional coefficient design is:

$$\alpha(t) = [\alpha_0(t), \alpha_1(t)] = [1 - t, t], \quad T = 1, \quad (19)$$

SI models $H_\theta(t, x(t)) = [H_{0,\theta}(t, x(t)), H_{1,\theta}(t, x(t))] = [\hat{x}_{0,\theta}, \hat{x}_{1,\theta}]$. The loss function is:

$$\mathcal{L}_{k,\theta} = \int_0^1 \mathbb{E}_{x_0,x_1}\left[|H_{k,\theta}(t, x(t))|^2 - 2x_k \cdot H_{k,\theta}(t, x(t))\right] dt, \quad k = 0, 1, \quad t \sim \mathcal{U}(0, 1). \quad (20)$$

## B. Justification for Diagonal Matrices in MAC

We argue that using diagonal matrices is sufficient to represent non-linear trajectories for inference trajectory optimization. Theoretically, the expressible space of the interpolated value $x(t)$ remains $x(t) \in \mathbb{R}^d$ for both the Hadamard product and matrix-vector multiplication. While the primary benefit of full matrix multiplication lies in its ability to model non-linear trajectories by incorporating cross-dimensional effects, we contend that a diagonal matrix is already adequate for this purpose—provided that the U-Net $U_\phi$, used to parameterize $\gamma_\phi$, possesses enough expressive power to generate diverse non-linear trajectories.

Moreover, using diagonal matrices for the output of MAC significantly reduces model size compared to using full matrices. One alternative is to use a low-rank-plus-diagonal decomposition, $\gamma_\phi = D + UV^T$, where $D = \gamma$ is a diagonal matrix, and $U, V \in \mathbb{R}^{d \times r}$, with $r$ being a typically small rank (e.g., $r = 4$ or $8$). However, even with a small $r$, this approach would still require a large number of output channels from $U_\phi$, significantly increasing model complexity. Specifically, $U_\phi$ would require an output channel size of $C \times (2r + 1) \times M \times 2$, where $C$ is the image channel size and $M$ is the number of sinusoidal functions. For instance, with $r = 4$, this results in $3 \times 9 \times 5 \times 2 = 270$ output channels for $U_\phi$ when $NFE = 5$, compared to only $C \times M \times 2 = 30$ channels in the diagonal case.

In conclusion, we believe that using a diagonal matrix, in conjunction with a sufficiently expressive U-Net $U_\phi$, provides ample capacity to model non-linear trajectories while keeping the model size efficient and manageable.

## C. Hypothesis Set Design for MAC

We design the hypothesis set and parameterization for $\gamma_\phi$ as follows:

$$
\Gamma_h = \left\{
\begin{aligned}
&\gamma_\phi(t, x_T) = [\gamma_{0,\phi}(t, x_T), \gamma_{1,\phi}(t, x_T)] : [0, T] \times \mathbb{R}^d \to \mathbb{R}^{d \times 2}, \quad \text{where} \\
&\gamma_{0,\phi}(t, x_T) = \mathcal{F}_0(b_m(t), w_\phi(x_T)) = T \frac{f_\phi(t, x_T)}{f_\phi(t, x_T) + g_\phi(t, x_T)}, \\
&\gamma_{1,\phi}(t, x_T) = \mathcal{F}_1(b_m(t), w_\phi(x_T)) = T \frac{g_\phi(t, x_T)}{f_\phi(t, x_T) + g_\phi(t, x_T)}, \quad \phi \in \mathcal{P}
\end{aligned}
\right\},
\tag{21}
$$

This parameterization can vary depending on the flow and diffusion framework. For example, we use $\gamma_{0,\phi}$ as described above for SI, but set $\gamma_{0,\phi}(t, x_T) = \mathbf{1}_d$ for EDM to align with its original formulation. The functions $f_\phi$ and $g_\phi$ are defined as:

$$
f_\phi(t, x_T) = 1 - \frac{t}{T} + \left( \sum_{m=1}^{M} w^f_{m,\phi}(x_T) b_m(t) \right)^2, \quad g_\phi(t, x_T) = \frac{t}{T} + \left( \sum_{m=1}^{M} w^g_{m,\phi}(x_T) b_m(t) \right)^2,
\tag{22}
$$

$$
b_m(t) = \sin\left( \pi m \left( \frac{t}{T} \right)^{1/q} \right) \in \mathbb{R}, \quad w_\phi(x_T) = s \, \text{LPF} \circ \tanh\left( \text{NN}_\phi(x_T) \right), \quad q \in \mathbb{R}, \quad s \in \mathbb{R},
\tag{23}
$$

where $M$ and $q$ are hyperparameters for the sinusoidal basis, and the scaling factor $s$ controls the output range of $w_\phi(x_T) \in (-s, s)$. When $s = 0.0$, $\gamma$ reduces to $\alpha$. For image generation, we use a U-Net $U_\phi$ as $\text{NN}_\phi$. The outputs $U_\phi(x_T) = [U_{f,\phi}(x_T), U_{g,\phi}(x_T)] \in \mathbb{R}^{C \times H \times W \times M \times 2}$, where $C$, $H$, and $W$ denote the channel, height, and width of the image. $U_{f,\phi}(x_T)$ and $U_{g,\phi}(x_T)$ each have a channel shape of $C \times M$, structured along the channel axis.

**Design Choice of Sinusoidals**    The inference time schedule of EDM is defined as:

$$
t_i = \left( t_{\max}^{\frac{1}{q}} + \frac{i}{N-1} \left( t_{\min}^{\frac{1}{q}} - t_{\max}^{\frac{1}{q}} \right) \right)^q, \quad t_{\min} = 0.002, \; t_{\max} = 80, \; q = 7.
\tag{24}
$$

As illustrated in Figure 9, $b_m(t) = \sin(\pi m (t/T)^{1/7})$ effectively covers the entire EDM time schedule, whereas $b_m(t) = \sin(\pi m (t/T))$ has minimal magnitude when $t \leq 1$. Since $w_\phi$ is constrained to $(-s, s)$, the choice of sinusoidal basis significantly influences the controllability of $\gamma_\phi$ during simulation. Accordingly, we use $q = 1$ for DDPM, FM, and SI, and $q = 7$ for EDM.

In addition, to avoid aliasing in $b_m(t)$, we follow the Nyquist criterion, requiring the sampling rate $f_s = N \geq 2f_{\max} = M$. Based on this, we set $M = N$ during adversarial training to ensure sufficient frequency resolution.

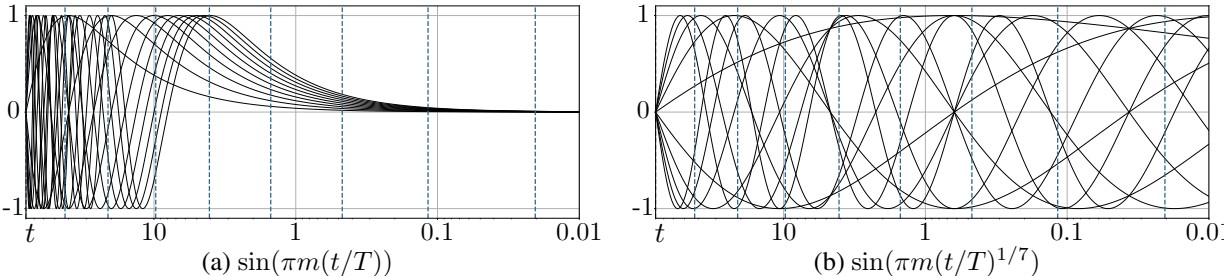

Figure 9: Comparison of $b_m(t)$ for $M = 10$. The dotted line represents the EDM inference time schedule for $N = 10$.

**Low-Pass Filtering (LPF)** For LPF, we apply a 2D convolution with a Gaussian kernel, where the kernel size is $\frac{20 \times \text{resolution}}{32} - 1$ and the standard deviation of the Gaussian is $\sigma = \frac{4.0 \times \text{resolution}}{32}$. Here, resolution refers to the image height or width. To mitigate boundary effects introduced by LPF, we apply zero-padding of $\frac{\text{kernel size}+1}{2}$ on all sides of $U_\phi$'s input and crop the borders after convolution to restore the original shape. To preserve consistent scaling, we compute the minimum and maximum values before LPF for each batch and rescale the post-LPF output to match the original range. LPF is applied only for image generation experiments.

**Hypothesis Set for Pre-training and Adversarial Training** Since using a large hypothesis set of coefficients during pre-training may overburden $H_\theta$ and degrade performance, we adopt a smaller hypothesis set for pre-training and enable full multidimensionality across $t$ only during adversarial training. For Section 4.2, LPF is used exclusively during pre-training and omitted during adversarial training. For Section 4.3, we configure the convolution group size for LPF during pre-training to 1, yielding an LPF output shape of $[B, 1, \text{resolution}, \text{resolution}]$. This enforces high multidimensionality for small $t$ and low multidimensionality for large $t$. In contrast, during adversarial training, we use a convolution group size of $[B, 2 \times 3 \times M, \text{resolution}, \text{resolution}]$, producing an LPF output shape of $[B, 2 \times 3 \times M, \text{resolution}, \text{resolution}]$.

# D. Details of the Differential Equation Solver

The displacement of the trajectory $x(t_{i+1}) - x(t_i)$, parameterized by $v_{\theta,\phi}$, is expressed as:

$$
\begin{aligned}
\Delta t_i \, v_{\theta,\phi}(t_i, x(t_i), x_T) &= \Delta t_i \, \dot{\gamma}_{0,\phi}(t_i, x_T) \odot \hat{x}_{0,\theta} + \Delta t_i \, \dot{\gamma}_{1,\phi}(t_i, x_T) \odot \hat{x}_{1,\theta} \\
&\approx \Delta \gamma_{0,\phi}(t_i, x_T) \odot \hat{x}_{0,\theta} + \Delta \gamma_{1,\phi}(t_i, x_T) \odot \hat{x}_{1,\theta},
\end{aligned}
\tag{25}
$$

where the time displacement $\Delta t_i = t_{i+1} - t_i$ is derived from the inference time schedule $\tau = \{t_0, \ldots, t_N\}$, with $t_0 = T > \ldots > t_N = 0$. The corresponding displacements for $\gamma$ are:

$$
\Delta \gamma_{0,\phi}(t_i, x_T) = \gamma_{0,\phi}(t_{i+1}, x_T) - \gamma_{0,\phi}(t_i, x_T), \quad \Delta \gamma_{1,\phi}(t_i, x_T) = \gamma_{1,\phi}(t_{i+1}, x_T) - \gamma_{1,\phi}(t_i, x_T).
\tag{26}
$$

This approach helps reduce numerical errors when solving differential equations under curved $\gamma$. Accordingly, the trajectory displacement for EDM is given by:

$$
\Delta t_i \, v_{\theta,\phi}(t_i, x(t_i), x_T) = \begin{cases} \dfrac{\Delta \gamma_{1,\phi}(t_i, x_T)}{\gamma_{1,\phi}(t_i, x_T)} \odot (x(t_i) - H_\theta(t_i, x(t_i), \gamma_\phi(t_i, x_T))) & \text{with } \gamma\text{-pre-training} \\[3ex] \dfrac{\Delta \gamma_{1,\phi}(t_i, x_T)}{\gamma_{1,\phi}(t_i, x_T)} \odot (x(t_i) - H_\theta(\bar{\gamma}_\phi(t_i, x_T), x(t_i))) & \text{w/o } \gamma\text{-pre-training} \end{cases}.
\tag{27}
$$

For SI:

$$
\Delta t_i \, v_{\theta,\phi}(t_i, x(t_i), x_T) = \Delta \gamma_{0,\phi}(t_i, x_T) \odot \hat{x}_{0,\theta} + \Delta \gamma_{1,\phi}(t_i, x_T) \odot \hat{x}_{1,\theta},
\tag{28}
$$

$$
[\hat{x}_{0,\theta}, \hat{x}_{1,\theta}] = \begin{cases} H_\theta(t_i, x(t_i), \gamma_\phi(t_i, x_T)) & \text{with } \gamma\text{-pre-training} \\ H_\theta(\bar{\gamma}_\phi(t_i, x_T), x(t_i)) & \text{w/o } \gamma\text{-pre-training} \end{cases}.
\tag{29}
$$

The differential equation solver $G_{\theta,\phi}$, using Euler discretization, is then defined as:

$$
G_{\theta,\phi}(\tau, x_T, v_{\theta,\phi}) = x_{\theta,\phi}(t_N) = x_T + \sum_{i=0}^{N-1} \Delta t_i \, v_{\theta,\phi}(t_i, x_{\theta,\phi}(t_i), x_T),
\tag{30}
$$

$$
x_{\theta,\phi}(t_{i+1}) \leftarrow x_{\theta,\phi}(t_i) + \Delta t_i \, v_{\theta,\phi}(t_i, x_{\theta,\phi}(t_i), x_T).
$$

# E. Details for Experiments

All experiments are conducted on NVIDIA RTX 3080Ti, RTX 4090, and RTX A6000 GPUs. Euler discretization is used for all inference procedures.

## E.1. 2-Dimensional Transportation: Optimizing Only $\phi$

### E.1.1. $\gamma$-Pre-Training

We follow the implementation from Tong et al. (2024), employing a multilayer perceptron (MLP) consisting of four linear layers with 64 hidden units and SiLU activation functions. The Stochastic Interpolant (SI) model is trained with a batch size of 256 for 20,000 iterations. The loss function is defined as:

$$\mathcal{L}_k(\theta) = \int_0^1 \mathbb{E}_{x_0, x_1, u} \left[ |H_{k,\theta}(t, x(t), \gamma(t, u))|^2 - 2x_k \cdot H_{k,\theta}(t, x(t), \gamma(t, u)) \right] dt, \ \ k = 0, 1, \ \ t \sim \mathcal{U}(0, 1), \ \ u \sim \mathcal{U}(-1, 1) \in \mathbb{R}^{d \times M \times 2}. \quad (31)$$

### E.1.2. Optimization of $\phi$

To train $\gamma_\phi$, we use a batch size of 1024 and run for 2000 iterations with $s = 0.1$. Each configuration is trained with three different random seeds, and we report the mean and standard deviation of the Wasserstein distance.

## E.2. Image Generation: Optimizing Only $\phi$

### E.2.1. Training Configurations

Table 8: U-Net configurations for $H_\theta$.

| Configuration | CIFAR-10 | ImageNet-32 |
|---|---|---|
| Channels | 128 | 256 |
| Depth | 2 | 3 |
| Channel multipliers | 1,2,2,2 | 1,2,2,2 |
| Attention heads | 4 | 4 |
| Head channel size | 64 | 64 |
| Attention resolution | 16 | 16 |
| Dropout | 0.1 | 0.1 |

We use the U-Net architecture from Dhariwal & Nichol (2021) for $H_\theta$ and the U-Net from Ronneberger et al. (2015) for $U_\phi$. Configuration details for $H_\theta$ are given in Table 8. For $\gamma_\phi$, we use a channel progression of [256, 512, 1024, 2048]. The discriminator $D_\psi$ consists of four convolutional layers with 1024 channels, batch normalization, leaky ReLU activations, and a final sigmoid output.

Table 9: Hyperparameters for pre-training and adversarial training.

| Hyperparameter | CIFAR-10 | | ImageNet-32 | |
|---|---|---|---|---|
| | Pre-training | Adversarial training | Pre-training | Adversarial training |
| Batch size | 128 | 16 | 512 | 15 |
| GPUs | 1 | 1 | 4 | 1 |
| Iterations | 400k | 200k | 250k | 200k |
| Peak LR | 2e-4 | 2e-4 | 2e-4 | 2e-4 |
| LR Scheduler | Poly decay | Poly decay | Poly decay | Poly decay |
| Warmup steps | 5k | 5k | 5k | 5k |
| Warmup steps for $D_\psi$ | – | 20k | – | 20k |

We use the Adam optimizer with $\beta_1 = 0.9$, $\beta_2 = 0.999$, weight decay of 0.0, and $\epsilon = 1 \times 10^{-8}$, along with polynomial decay learning rate scheduling. An exponential moving average (EMA) with a decay rate of 0.999 is applied during all training phases. For adversarial training, we use the vanilla GAN loss from Equation 5 instead of hinge loss for simplicity. FID is evaluated every 10,000 steps, and we report the lowest value obtained. Hyperparameters are summarized in Table 9.

E.2.2. EXPERIMENTS FOR HYPOTHESIS SET HYPERPARAMETER TUNING

Table 10: FID comparison between $\alpha$ and $\gamma$ under various configurations (e.g., $s$, LPF) across different NFE.

| Method \ NFE | CIFAR-10 | | | | ImageNet-32 | | | |
|---|---|---|---|---|---|---|---|---|
| | 10 | 100 | 150 | 200 | 10 | 100 | 150 | 200 |
| $\text{SI}_\alpha$ | 14.43 | 4.75 | 4.51 | 4.30 | 17.72 | 8.08 | 7.79 | 7.63 |
| $\text{SI}_{\gamma(s=0.005)}$ | 14.59 | 3.98 | 3.74 | 3.63 | 17.41 | 6.33 | 6.21 | 6.20 |
| $\text{SI}_{\gamma(s=0.1, \text{LPF})}$ | 15.44 | 3.77 | 3.68 | 3.75 | 17.86 | 6.63 | 6.47 | 6.44 |
| $\text{FM}_\alpha$ | 13.70 | 4.52 | 4.23 | 4.07 | 16.92 | 7.78 | 7.53 | 7.38 |
| $\text{FM}_{\gamma(s=0.005)}$ | 13.81 | 3.59 | 3.42 | 3.42 | 16.85 | 6.18 | 6.03 | 6.01 |
| $\text{FM}_{\gamma(s=0.1, \text{LPF})}$ | 15.13 | 3.64 | 3.57 | 3.64 | 17.52 | 6.40 | 6.27 | 6.31 |
| $\text{DDPM}_\alpha$ | 98.47 | 6.64 | 4.84 | 4.10 | 111.54 | 8.13 | 7.40 | 7.14 |
| $\text{DDPM}_{\gamma(s=0.005)}$ | 74.44 | 3.77 | 5.96 | 7.84 | 139.69 | 7.67 | 12.37 | 11.70 |
| $\text{DDPM}_{\gamma(s=0.005,\text{LPF})}$ | 72.23 | 4.73 | 4.11 | 3.83 | 135.48 | 6.84 | 6.51 | 6.42 |
| $\text{DDPM}_{\gamma(s=0.1, \text{LPF})}$ | 71.80 | 4.46 | 6.32 | 12.60 | 142.99 | 6.70 | 8.69 | 10.91 |

Table 11: FID for different NFE and Gaussian kernel $\sigma$ in LPF, evaluated on CIFAR-10 with $\text{SI}_{\gamma(s=0.1,\text{LPF})}$.

| $\sigma$ \ NFE | 10 | 20 | 30 | 40 | 50 | 100 | 150 | 200 |
|---|---|---|---|---|---|---|---|---|
| 0.1 | 14.89 | 8.05 | 6.49 | 5.45 | 6.53 | 9.59 | 10.67 | 11.20 |
| 1.0 | 14.92 | 8.47 | 5.32 | 4.45 | 4.68 | 6.06 | 7.01 | 7.50 |
| 2.0 | 16.25 | 9.56 | 7.56 | 6.06 | 4.72 | 3.77 | 3.95 | 4.17 |
| 4.0 | 15.44 | 9.13 | 7.39 | 6.36 | 5.59 | 3.77 | 3.68 | 3.75 |

Table 12: FID after adversarial training with 10 NFE on CIFAR-10 for varying values of $M$.

| Method \ $M$ | 5 | 10 | 15 | 20 | 25 | 30 |
|---|---|---|---|---|---|---|
| $\text{SI}_{\gamma(s=0.1, \text{LPF})}$ | 6.89 | 4.14 | 4.42 | 5.32 | 6.11 | 5.74 |
| $\text{FM}_{\gamma(s=0.1, \text{LPF})}$ | 5.93 | 6.13 | 6.70 | 6.18 | 5.97 | 6.42 |
| $\text{DDPM}_{\gamma(s=0.1, \text{LPF})}$ | 10.15 | 10.04 | 9.60 | 9.04 | 8.94 | 9.19 |

Table 13: FID after adversarial training of SI under various configurations on CIFAR-10 with 10 NFE.

| Method \ $M$ | 5 | 10 | 15 | 20 |
|---|---|---|---|---|
| $s = 0.0$ | 10.20 | 9.75 | 11.30 | 11.53 |
| $s = 0.005$ | 6.60 | 4.79 | 4.45 | 5.28 |
| $s = 0.005$, LPF | 7.37 | 4.42 | 4.26 | 5.31 |
| $s = 0.1$, LPF | 7.21 | 4.14 | 5.59 | 5.32 |

We conduct extensive experiments to tune hyperparameters for the hypothesis set. Tables 10 and 11 show the effects of different coefficients, LPF configurations, and other settings during pre-training. Tables 12 and 13 present results from adversarial training under various configurations. These results help identify optimal hyperparameter settings that enhance model performance in both pre-training and adversarial training stages.

## E.3. Image Generation: Optimizing $\theta$ and $\phi$

### E.3.1. $\gamma$-PRE-TRAINING

Table 14: Hyperparameters used for pre-training EDM.

| Hyperparameter | CIFAR-10 | FFHQ & AFHQ |
|---|---|---|
| Number of GPUs | 8 | 8 |
| Duration (Mimg) | 200 | 200 |
| Minibatch size | 512 | 256 |
| Learning rate | 1e-3 | 2e-4 |
| LR ramp-up (Mimg) | 10 | 10 |
| EMA half-life (Mimg) | 0.5 | 0.5 |
| Dropout probability | 13% | 5% (FFHQ) / 25% (AFHQ) |
| Channel multiplier | 128 | 128 |
| Channels per resolution | 2-2-2 | 1-2-2-2 |
| Augment probability | 12% | 15% |
| $M$ | 10 | 10 |
| Low-pass filtering | True | True |
| $s$ | 0.05 | 0.05 |

We adopt the codebase from Karras et al. (2022) and follow the EDM configuration, replacing $\alpha$ with $\gamma$ and applying coefficient conditioning. The pre-training loss function is:

$$\mathcal{L}_\theta = \mathbb{E}_{t,x_0,x_1,u} \left[ \lambda(t,u) c_{\text{out}}(t,u)^2 \left\| F_\theta\left(c_{\text{noise}}(t), c_{\text{in}}(t,u)x(t), c_{\text{coeff}}(t,u)\right) - \frac{1}{c_{\text{out}}(t,u)} \left(x_0 - c_{\text{skip}}(t,u)x(t)\right) \right\|_2^2 \right], \quad (32)$$

where:

$$c_{\text{in}}(t,u) = \frac{1}{\sqrt{\gamma_1^2(t,u) + \sigma_{\text{data}}^2}}, \quad c_{\text{out}}(t,u) = \frac{\gamma_1(t,u) \cdot \sigma_{\text{data}}}{\sqrt{\sigma_{\text{data}}^2 + \gamma_1^2(t,u)}}, \quad c_{\text{skip}}(t,u) = \frac{\sigma_{\text{data}}^2}{\gamma_1^2(t,u) + \sigma_{\text{data}}^2},$$

$$c_{\text{noise}}(t) = \frac{1}{4}\ln t, \quad c_{\text{coeff}}(t,u) = \frac{1}{4}\ln\gamma_1(t,u), \quad \lambda(t,u) = \frac{\gamma_1^2(t,u) + \sigma_{\text{data}}^2}{(\gamma_1(t,u) \cdot \sigma_{\text{data}})^2}, \quad (33)$$

with $\ln(t) \sim \mathcal{N}(-1.2, 1.2^2)$, $u \sim \mathcal{N}(-1,1) \in \mathbb{R}^{d \times M \times 2}$, and $\sigma_{\text{data}} = 0.5$. For coefficient conditioning (Section 3.2.2), we concatenate $[c_{\text{in}}(t,u)x(t), c_{\text{coeff}}(t,u)]$ along the channel axis as input to $F_\theta$. We use the Adam optimizer with $\beta_1 = 0.9$, $\beta_2 = 0.999$, and $\epsilon = 1e-8$.

### E.3.2. ADVERSARIAL TRAINING

Table 15: Hyperparameters used for adversarial training.

| Hyperparameter | CIFAR-10 | FFHQ & AFHQ | ImageNet |
|---|---|---|---|
| Number of GPUs | 8 | 8 | 8 |
| Training duration for $\gamma_\phi$ (kimg) | 1500 | 1000 | 30 |
| Minibatch size for $v_\theta$ | 512 | 256 | 256 |
| Minibatch size for $\gamma_\phi$ | 128 | 64 | 32 |
| Minibatch size for $D_\psi$ | 128 | 64 | 512 |
| Learning rate for $v_\theta$ | 1e-5 | 1e-5 | 1e-5 |
| Learning rate for $\gamma_\phi$ | 1e-4 | 1e-4 | 1e-4 |
| Learning rate for $D_\psi$ | 1e-3 | 1e-3 | 1e-3 |
| EMA half-life (kimg) | 10 | 10 | 10 |
| Low-pass filtering | True | True | True |
| $s$ | 0.05 | 0.05 | 0.05 |

For $U_\phi$, we adopt a U-Net architecture based on Song et al. (2021) with the following configuration: 256 channels, channel multipliers [1, 2, 4], a dimensionality multiplier of 4, 4 blocks, and an attention resolution of 16. The embedding layer for $t$ is disabled. We disable dropout in both $H_\theta$ and $\gamma_\phi$ to ensure deterministic behavior of $G_{\theta,\phi}$. We use the Adam optimizer with $\beta_1 = 0.0$, $\beta_2 = 0.99$, and $\epsilon = 1e{-}8$. The sampling time $t$ is drawn from $\ln(t) \sim \mathcal{N}(-1.2, 1.2^2)$ and is quantized according to the inference time schedule $\tau$. For ablation studies, each configuration is trained for 500 kimg, corresponding to approximately 4000 iterations. When training only $\phi$ (Table 6, Figure 7), LPF is not applied.

### E.3.3. OPTIMIZED INFERENCE TRAJECTORIES DEVIATE FROM LINEARITY

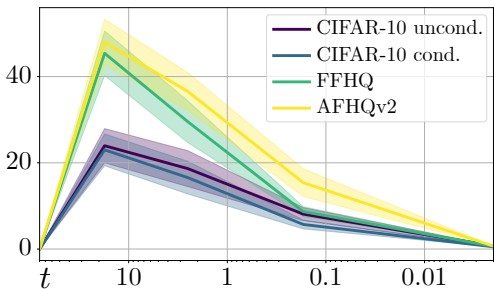

Figure 10: $L_2$ norm between the linear trajectory $x_{\mathrm{lin}}(t)$ and the optimized trajectory $x_{\theta,\phi}(t)$ over $t$.

To confirm that the optimized inference trajectories in the image generation experiment are not linear lines, we compute the $L_2$ norm between linear trajectories $x_{\mathrm{lin}}(t) = \frac{t}{T}x_T + \left(1 - \frac{t}{T}\right)x_{\theta,\phi}(t_N)$ and the optimized trajectories $x_{\theta,\phi}(t)$. As shown in Figure 10, the optimized inference trajectories clearly deviate from the linear path. This result demonstrates that our method effectively discovers superior, non-linear trajectories in high-dimensional spaces, leading to enhanced performance.

## F. Metrics Calculation

For CIFAR-10, AFHQ, and FFHQ experiments, we follow the evaluation protocol and code provided by Karras et al. (2022) to compute the Fréchet Inception Distance (FID). For ImageNet-64 experiments, we use the ImageNet dataset from EDM (Karras et al., 2022) as the reference and calculate both FID and $FD_{\mathrm{DINOv2}}$ using the code from EDM2 (Karras et al., 2024). All metrics are computed over 50,000 generated samples. Each experiment is run three times with different random seeds, and we report the minimum FID and $FD_{\mathrm{DINOv2}}$ values observed.

## G. Generated Samples

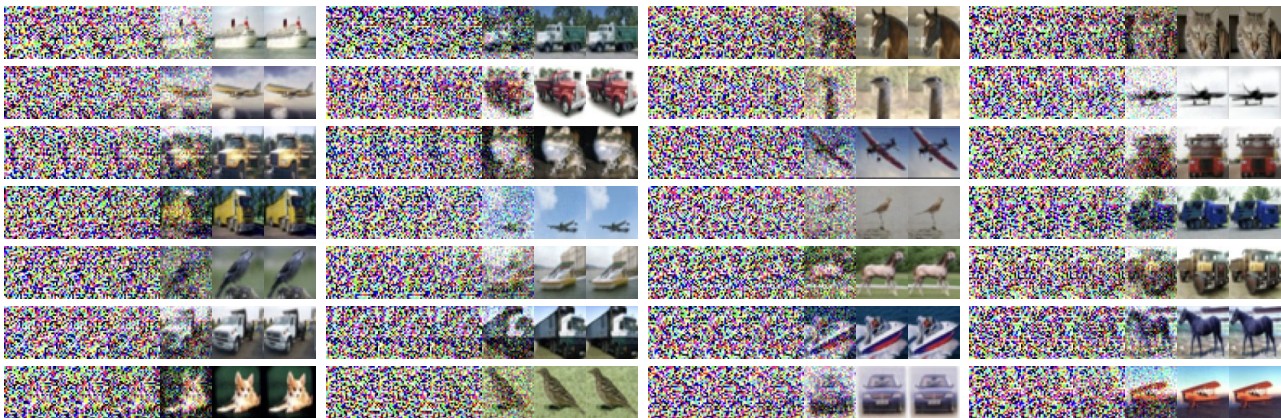

Figure 11: Inference trajectories from $\mathrm{EDM}_\gamma$ + adv $\theta, \phi$ for conditional generation on CIFAR-10.

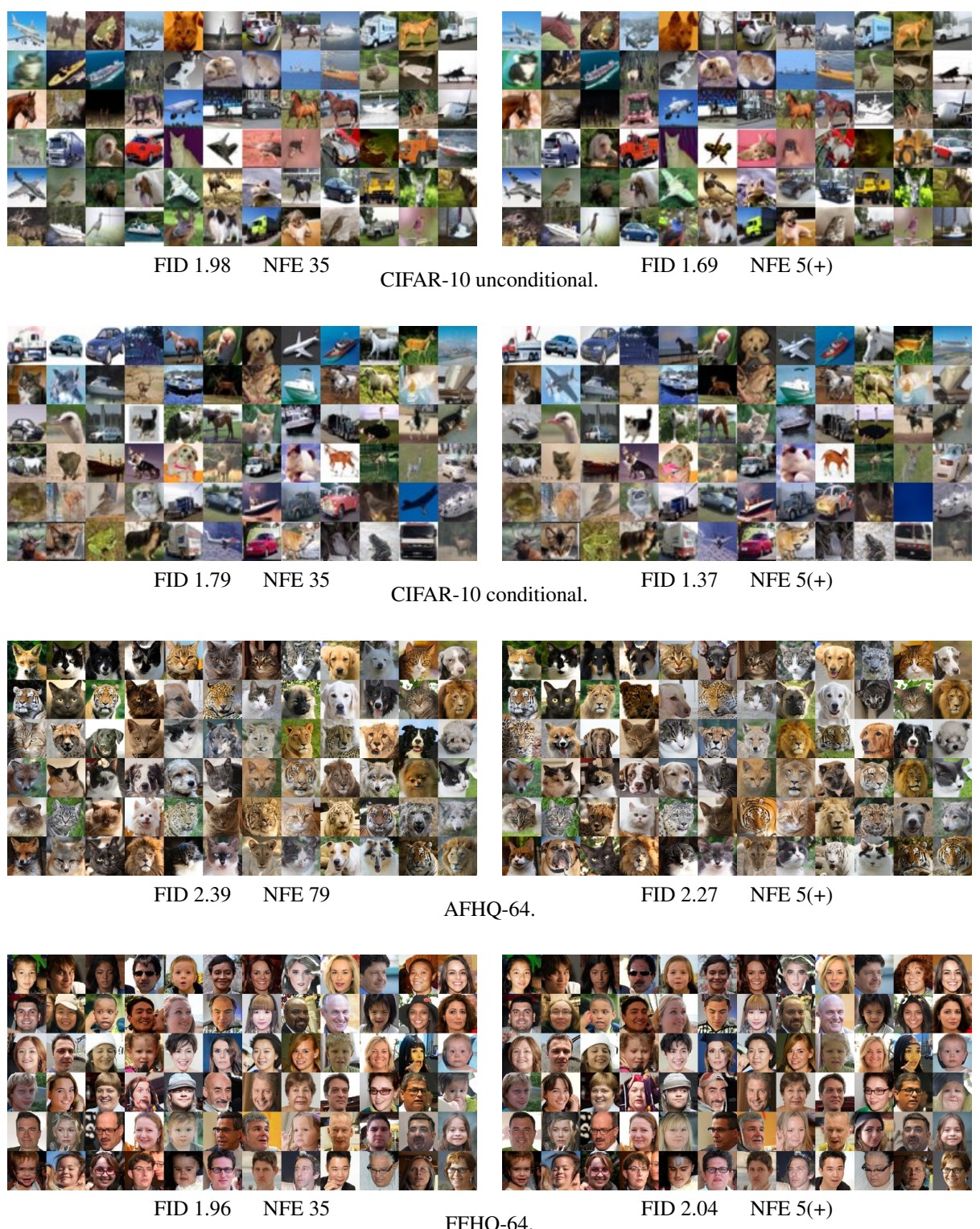

FID 1.98     NFE 35      CIFAR-10 unconditional.      FID 1.69     NFE 5(+)

FID 1.79     NFE 35      CIFAR-10 conditional.       FID 1.37     NFE 5(+)

FID 2.39     NFE 79      AFHQ-64.                    FID 2.27     NFE 5(+)

FID 1.96     NFE 35      FFHQ-64.                    FID 2.04     NFE 5(+)

Figure 12: Generated samples on CIFAR-10, FFHQ-64, and AFHQ-64 using $\mathrm{EDM}_\alpha$ (left) and $\mathrm{EDM}_\gamma$ + adv $\theta, \phi$ (right).

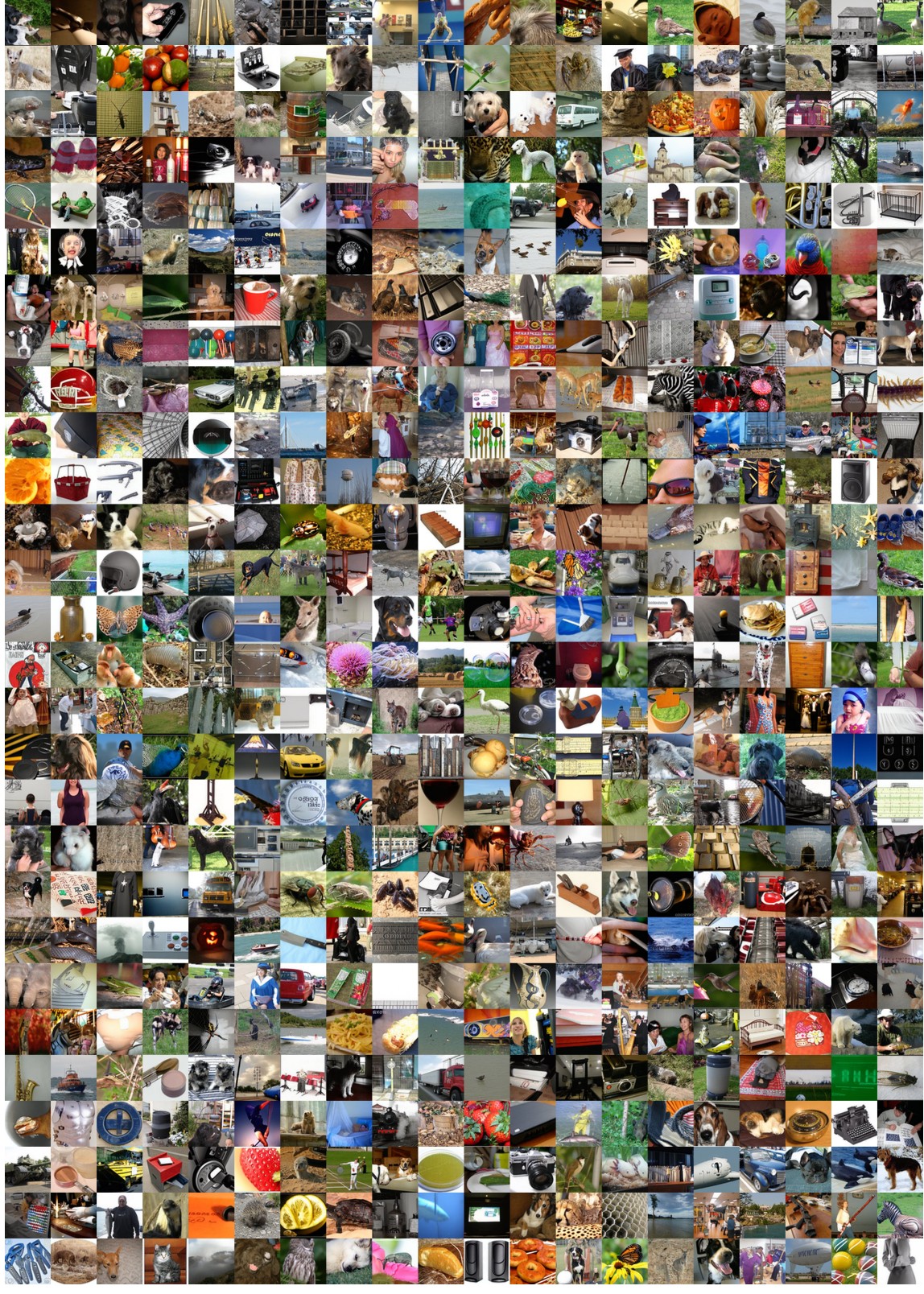

FID 1.48     FD$_{\text{DINOv2}}$ 70.2     NFE 5(+)

Figure 13: Generated samples on ImageNet-64 using EDM$_\alpha$ + adv $\theta, \phi$.

