# OpenReview forum: "Multidimensional Adaptive Coefficient for Inference Trajectory Optimization in Flow and Diffusion"
_ICML.cc/2025/Conference — ICML 2025 poster_

### Official Review · Reviewer_jRK3 · 2025-03-02

**Overall Recommendation:** 3

**Summary:**

This paper proposes to optimize model parameters and forward process interpolation coefficients with respect to a simulation-based adversarial loss. The training process consists of two stages. In the first stage, the flow model is trained w.r.t. randomly sampled multi-dimensional interpolation schemes via flow matching. To constrain the hypothesis space of interpolation schemes, interpolation coefficients are parameterized as a sum of weighted sinusoidals. In the second stage, the flow model and interpolation coefficients are optimized to minimize a simulation-based adversarial loss. Here, the weights for sinusoidals are parameterized as neural net outputs conditioned on $x_T$, providing an additional level of adaptivity. The authors demonstrate that the proposed method achieves competitive FID scores on CIFAR-10, ImageNet $32 \times 32$, FFHQ $64 \times 64$, and AFHQ $64 \times 64$.

**Claims And Evidence:**

- **(Left column, lines 260-262) "Given that $H_\theta(t, x(t)) \approx x_0$, $H_\theta$ can be adversarially refined by optimizing it with the discriminator $D_\psi$."**

This claim holds only when generative paths are sufficiently straight. However, as cited below, the authors also claim that the benefit of using MAC arises from its ability to discover better non-linear trajectories.

> (Left column, lines 300-305) "This suggests that a straight trajectory is not always optimal, even in OT-trained models, and MAC can adaptively discover better trajectories to correct errors that arise during transportation. Figure 3 further illustrates how MAC adjusts the trajectory direction to optimize transportation, resulting in a path that is not straight."

Hence, the authors' assumption $H_\theta(t, x(t)) \approx x_0$ that justifies the usage of adversarial learning on velocity $H_\theta$ output instead of generator $G_{\theta,\phi}$ output seems to be at odds with the benefits of using MAC.

**Essential References Not Discussed:**

This paper is missing a discussion of [1,2,3]. [1] also optimizes the forward process to learn faster flows. [2,3] also incorporates adversarial learning into flow matching for improved generative modeling.

[1] Minimizing Trajectory Curvature of ODE-based Generative Models, ICML, 2023

[2] Constant Acceleration Flow, NeurIPS, 2024

[3] Simple ReFlow: Improved Techniques for Fast Flow Models, ICLR, 2025

**Experimental Designs Or Analyses:**

I did not find any issues with experimental designs or analysis.

**Methods And Evaluation Criteria:**

CIFAR-10, FFHQ-64, AFHQ-64 are standard datasets / FID and IS are standard metrics for evaluating the performance of generative models. However, based on previous works such as [1,2,3], I believe results on at least ImageNet $64 \times 64$ are necessary to demonstrate the practicality and scalability of the proposed method.

[1] Consistency Trajectory Models: Learning Probability Flow ODE Trajectory of Diffusion, ICLR, 2024

[2] Constant Acceleration Flow, NeurIPS, 2024

[3] Simple ReFlow: Improved Techniques for Fast Flow Models, ICLR, 2025

**Other Comments Or Suggestions:**

- Right column, lines 43-48 : $x\_{est,\theta,\phi}$ is undefined.
- Right column, Eq. (2) : $\hat{x}\_{0,\theta}$ and $\hat{x}\_{1,\theta}$ are undefined.
- Table 4 : please clarify the difference between $z \sim \rho_T$ and $x_T \sim \rho_T$.

**Other Strengths And Weaknesses:**

- **originality and significance** : while the usage of adversarial learning to improve the perceptual quality of neural ODEs have been already introduced in works such as [1] or [2], the idea of optimizing forward process parameters via simulation-based training is novel and interesting. Still, I find the experimental setup rather limited, and as discussed in **Methods And Evaluation Criteria**, additional results on higher resolution or more complex datasets will significantly strengthen this submission.

[1] Consistency Trajectory Models: Learning Probability Flow ODE Trajectory of Diffusion, ICLR, 2024

[2] Constant Acceleration Flow, NeurIPS, 2024

**Questions For Authors:**

I am willing to raise the score to **Weak Accept** if the authors can address the following concerns.

- [Q1] Can the authors provide any clarifications regarding the comment in **Claims And Evidence**?
- [Q2] Can the authors provide additional experiments on ImageNet $64 \times 64$ as a demonstration of scalability?

**Relation To Broader Scientific Literature:**

This paper proposes to adversarially optimize forward process interpolation coefficients for better generative modeling.

**Theoretical Claims:**

There are no new theoretical claims.

---

> ### Author Rebuttal · Authors · 2025-04-01
>
> URL for Additional Figures: https://imgur.com/a/3UiYDVF
>
> $\textbf{[A1] $H_\theta$ estimates the vector field and $\gamma_\phi$ performs inference-time planning}$
>
> We clarify the misunderstanding here. As shown in Additional Figure 2, the roles of $\theta$ and $\phi$ differ clearly. $H_{\theta}$ estimates the vector field by predicting the endpoint $x_0$ regardless of trajectory curvature. Adaptive coefficients ($\gamma_{\phi}$) determine nonlinear trajectory planning by adjusting velocity directions and step sizes based on predictions from $H_{\theta}$. Thus, the capability for nonlinear trajectory planning depends entirely on $\gamma_{\phi}$, which adaptively guides the inference until endpoints are reached.
>
> For clarity, we added Additional Figure 1 to illustrate MAC enabling distinct, adaptively curved trajectories for inference planning. Our core contribution, MAC, provides inference-time planning, meaning optimal inference plans are computed and optimized offline via simulation and then directly deployed during inference without additional computations.
>
> $\textbf{[A2] Scalability Experiments on ImageNet}$
>
> We agree with the reviewer that experiments on ImageNet are essential for demonstrating scalability and practical value. However, due to substantial resource requirements (32 A100 GPUs for 2 weeks needed for pre-training $EDM_\gamma$ on ImageNet-64), conducting large-scale experiments within the rebuttal period is currently impractical.
>
> Nevertheless, we emphasize our method’s scalability, supported by the following points:
>
> As in current SoTA methods on ImageNet-256 ([5], [6], [7], [8]), diffusion models’ scalability primarily relies on latent diffusion paradigms [4]. Our ongoing research explicitly integrates MAC with LDM, discrete diffusion frameworks [7], and IMM [8]. These frameworks substantially reduce computational demands, making ImageNet-scale experiments feasible by leveraging available pre-trained models.
>
> Specifically, given that discrete diffusion frameworks naturally use highly multidimensional interpolated values for pre-training, optimizing MAC on discrete diffusion frameworks enables large-scale experiments using existing pre-trained models. Additionally, combining MAC with advanced models like IMM [8] can achieve high performance on large-scale datasets by using existing pre-trained models (explained in our response to Reviewer TFVT [A3]).
>
> MAC can be readily integrated with these frameworks, and we are currently preparing dedicated large-scale follow-up studies.
>
> For revision, we propose including experiments on ImageNet-64 using our existing setup with available $EDM_\alpha$ models. We believe this sufficiently demonstrates our method’s scalability within the scope of the original submission.
>
> $\textbf{[A3] Essential References Discussion}$
>
> Thank you for identifying the missing literature ([1], [2], [3]). Briefly, the major conceptual difference is that prior works ([1], [2], [3]) optimize vector fields by predefined trajectory properties (e.g., straightness or minimal curvature) guided by optimal transport theory to reduce numerical errors. In contrast, our method does not enforce predefined inference trajectory optimality; instead, we adaptively discover optimal nonlinear trajectories by final transportation quality through simulation-based inference-time planning. This allows greater flexibility and potentially superior performance.
>
> $\textbf{[A4] Notation Clarifications}$
>
> $G_{\theta, \phi}(\tau, x_T) = x_{\text{est}, \theta, \phi}$ (Details in Appendix C).
>
> $\hat{x}$ values are predictions from $H_{\theta}(t, x(t))$ for terminal points $x_0 \sim \rho_0$ and $x_1 \sim \rho_1$.
>
> $x_T$ denotes the actual initial sample point, while $z$ is sampled from the same distribution but unused during inference.
>
> We sincerely appreciate your thoughtful consideration and hope these clarifications effectively address your concerns.
>
> [1] Minimizing Trajectory Curvature of ODE-based Generative Models, ICML, 2023
>
> [2] Constant Acceleration Flow, NeurIPS, 2024
>
> [3] Simple ReFlow: Improved Techniques for Fast Flow Models, ICLR, 2025
>
> [4] High-resolution Image Synthesis with Latent Diffusion Models, CVPR, 2021
>
> [5] Scalable Diffusion Models with Transformers, ICCV, 2023
>
> [6] SiT: Exploring Flow and Diffusion-based Generative Models with Scalable Interpolant Transformers, ECCV, 2024
>
> [7] [Mask] is All You Need, arXiv, 2025
>
> [8] Inductive Moment Matching, arXiv, 2025

---

> > ### Comment · Reviewer_jRK3 · 2025-04-08
> >
> > Thank you for the reply. Given the authors rebuttal and additional experiments on ImageNet64, I have raised the score from **Weak Reject** to **Weak Accept**.

---

### Official Review · Reviewer_TFVT · 2025-03-10

**Overall Recommendation:** 3

**Summary:**

This paper introduces Multidimensional Adaptive Coefficients (MAC) for flow and diffusion models, allowing coefficients to vary across dimensions and adapt to different starting points. The two-stage approach combines pre-training with multidimensional coefficients and adversarial refinement. Experiments across multiple frameworks and datasets show improved performance, including state-of-the-art results on CIFAR-10 conditional generation.

## update after rebuttal
The author resolved my confusion. However, considering that reviewer MR8r's review was not addressed, I'm unsure what happened. Finally, considering the contribution of the current work, my current rating remains unchanged.

**Claims And Evidence:**

The performance improvement claims are generally supported by comprehensive experiments across multiple frameworks and datasets.

However, the "training efficiency" claim is problematic since the method requires training models from scratch rather than fine-tuning existing pre-trained models, which is a significant practical limitation.

**Essential References Not Discussed:**

I find that the existing citations sufficiently cover the related literature, though the novelty of the proposed approach remains limited.

**Experimental Designs Or Analyses:**

The experimental design is overall sound, with comprehensive testing across multiple frameworks and datasets.

The ablation study in Table 6 shows that adversarial training provides most of the performance gain. Since adversarial training is not a novel contribution of this paper, this suggests that MAC's specific contribution to performance improvement is relatively limited.

**Methods And Evaluation Criteria:**

The evaluation metrics and benchmark datasets are appropriate for the task.

However, there is a concern with the methodology. The hypothesis space design for MAC introduces many additional hyperparameters (M, s, q, LPF configuration). The paper lacks analysis of how baseline methods might perform if augmented with similar parameterization advantages. For example, it would be valuable to see how CTM would perform if pre-trained with comparable additional parameters before distillation.

**Other Comments Or Suggestions:**

The paper would benefit from providing a more detailed analysis of computational overhead. Additionally, testing on higher-resolution images would help demonstrate the method's scalability to more complex generation tasks.

**Other Strengths And Weaknesses:**

**Strengths**:

1. Comprehensive experiments across multiple frameworks demonstrate the method's versatility

2. Good visualizations of learned trajectories help explain the method's operation

3. The approach achieves state-of-the-art results on CIFAR-10 conditional generation

**Weaknesses**:

1. The requirement to retrain models from scratch makes the method impractical for large-scale applications

2. The performance gains attributable specifically to MAC (versus general adversarial refinement) appear modest

3. The method introduces significant additional complexity through MAC hyperparameters

**Questions For Authors:**

When considering the total computation (pre-training + adversarial finetuning), how does your method compare to these alternatives?

**Relation To Broader Scientific Literature:**

The paper positions itself as novel by optimizing trajectory quality through simulation, but this approach is conceptually similar to:

1. Kim et al. (2024) (CTM) which also uses adversarial refinement on diffusion trajectories
2. Lu et al. (2024) (CD + GAN) which likewise employs adversarial refinement

The primary difference, multidimensional coefficients, appears to provide only incremental benefits over standard adversarial refinement of model parameters.

**Theoretical Claims:**

There is no theoretical claims in this paper.

---

> ### Author Rebuttal · Authors · 2025-04-01
>
> URL for Additional Figures: https://imgur.com/a/3UiYDVF
>
> $\textbf{[A1] MAC’s core value lies in inference-time planning, orthogonal to vector field tuning methods like CTM and CD+GAN}$
>
> CTM and CD+GAN optimize trajectories by adjusting the vector field parameter $\theta$. In contrast, our method (MAC) performs inference-time planning, meaning the optimal inference plans are computed and optimized offline via simulation and then directly deployed during inference without additional computations.
>
> As shown in Additional Figures 1 and 2, without MAC, trajectory planning’s search scope is limited to linear paths, restricting performance gains. MAC enables adaptive, nonlinear, dimension-wise trajectory planning, significantly expanding optimization flexibility.
>
> CTM and CD+GAN do not employ actual inference-time simulation feedback. Our method dynamically optimizes trajectories and timestep plans using simulations identical to inference, making it uniquely dynamic.
>
> MAC acts as a final-stage inference planning strategy, enhancing performance beyond what is achievable by optimizing the vector field alone. This means that MAC can be integrated with existing frameworks such as CTM and other diffusion or distillation methods, as their methodologies attempt to solve different problems.
>
> $\textbf{[A2] MAC Reduces Hyperparameter Engineering Costs}$
>
> The modest gains observed with EDM result from EDM’s already highly optimized configuration, leaving limited margin for further improvement. (This is connected to the reason why we use $EDM_\gamma$ and coefficient labeling, explained below in [A3].) However, Sections 4.1 and 4.2 demonstrate significant improvements (~10x NFE efficiency) in less optimized frameworks (DDPM, FM, SI) by using MAC.
>
> MAC alleviates the extensive manual tuning burden associated with framework-specific hyperparameters, like in EDM. Although MAC requires tuning of hypothesis space parameters ($M, s, q$, LPF configs), these parameters are framework-independent. Hence, once identified, these parameters generalize well across various flow and diffusion models, providing considerable performance benefits without framework-specific engineering, as shown in our experiments.
>
> Regarding baseline comparison with similar parameterization: As demonstrated in Table 6, directly incorporating pre-training parameterization into EDM ($EDM_\gamma$) negatively impacts performance. Using this as a teacher network will degrade CTM performance unless combined with MAC for inference-time planning. MAC provides clear advantages specifically when used for adaptive inference-time planning, not as a general pre-training strategy.
>
> $\textbf{[A3] Practicality via Existing Pre-trained Models}$
>
> Our method indeed requires higher computational costs compared to CTM (approximately 201 Mimg vs. 25 Mimg) when using the specific $EDM_\gamma$ pre-training setup, chosen deliberately to push the limits and achieve SoTA performance on CIFAR-10. However, we emphasize that using $EDM_\gamma$ and coefficient labeling is optional. MAC can practically and effectively leverage existing pre-trained models (such as $EDM_\alpha$), significantly reducing computational overhead without substantially compromising performance.
>
> Using $\gamma$ increases the probability of multidimensional interpolated values $x(t)$, but even using $\alpha$ inherently yields multidimensional interpolations, as Gaussian noise is independently sampled across dimensions. Specifically, given $x_1 \sim \rho_1 = \mathcal{N}(0, I_d) \in \mathbb{R}^d$, each dimension $x_{1,i}$ is independently drawn from $\mathcal{N}(0,1)$, and thus even a linear interpolation $x(t) = t \cdot x_1 + (1-t) \cdot x_0$ naturally results in dimension-wise distinct noise contributions.
>
> Empirical evidence from our experiments supports this claim. Section 4.1 demonstrates effective nonlinear trajectory planning in $OT-SI_{MAC}$ using models pre-trained with $\alpha$. Additionally, Figure 6 shows the trained $\phi$‘s distribution significantly differs from the pre-training distribution, indicating MAC’s effectiveness is not critically dependent on pre-training specifics.
>
> Thus, for large-scale practical scenarios where training from scratch is costly, MAC can effectively utilize existing models (e.g., $EDM_\alpha$), ensuring practicality and computational efficiency.
>
> $\textbf{[A4] Scalability (ImageNet)}$
>
> Please refer to our response to Reviewer jRK3’s comment [A2].
>
> We sincerely appreciate your thoughtful consideration and hope these clarifications effectively address your concerns.

---

> > ### Comment · Reviewer_TFVT · 2025-04-02
> >
> > Thanks for the authors' responses. My final concern lies in complete experimental comparisons, just like all the other reviewers, i.e., experiments on ImageNet64. While the authors note that extensive GPU resources would be needed for these experiments, there are strategies available to reduce computational demands, such as gradient accumulation. Although a complete comparison would be time-intensive, the preliminary results produced by a not fully converged model, could provide meaningful insights into the method's scalability. Given this, I maintain my current score.

---

> > > ### Author Response · Authors · 2025-04-06
> > >
> > > $\textbf{[A] ImageNet-64 Result Demonstrating Scalability and Practicality Using Only Existing Pre-trained Models}$
> > >
> > > We would like to inform the reviewers that we have conducted an additional experiment on ImageNet-64, as requested.
> > >
> > > We achieved an FID of $\textbf{1.47}$ with 5(+) NFE by applying inference trajectory optimization $\textbf{using an existing pre-trained}$ $EDM_\alpha$ model, with only $\textbf{30k}$ training images for MAC. This result outperforms CTM with NFE = 2 (FID = 1.73), which requires $\textbf{61.4M}$ training images. Except for the batch size, which was set to 32 for $\gamma_\phi$, and 512 for $D_\psi$, all other configurations—including the model size for $w_\phi$—were kept identical to those used in the CIFAR-10, FFHQ, and AFHQ experiments.
> > >
> > > This demonstrates a substantial gain in training efficiency and scalability—achieving better performance with $\textbf{2048}\times$ fewer training samples than CTM. Furthermore, this result empirically supports our earlier rebuttal point [A3] to Reviewer TFVT, as it was obtained $\textbf{without any pre-training stage}$, directly proceeding to the adversarial trajectory optimization stage using an existing pre-trained model without any modification.
> > >
> > > We note that further training may improve FID scores, but we report the result as early as possible within the rebuttal period to provide a timely response. Additional ablation studies on ImageNet-64 will be included in the revision.
> > >
> > > We hope this new result effectively addresses the concerns regarding the practicality and scalability of our method.

---

### Official Review · Reviewer_MR8r · 2025-03-14

**Overall Recommendation:** 2

**Summary:**

This paper introduces a new way to handle the interpolation between noise and data in diffusion and flow-based generative models. Unlike standard approaches that use an interpolation with a uniform scale across the entire image (like Rectified Flows, DDPM, and IDDPM), the authors propose extending this interpolation to a 2D space, matching the dimensions of the image and noise. This 2D interpolation is learned using a coefficient model, offering potentially more flexibility in the interpolation process. The training procedure involves three main components: a standard diffusion model trained with a 2D sampling of the time variable, a discriminator, and the coefficient model, all optimized jointly in the second stage to find better interpolation weights.

## update after rebuttal

As I didn't receive any rebuttal from the author I'll remain my current rating unchanged.

After reviewing the author's rebuttal with other reviewers, I believe most of my concerns remain unaddressed, and the paper, especially the experiment section, still appears incomplete and not yet ready from my perspective.

**Claims And Evidence:**

The central claim is that the proposed 2D learned interpolation offers more flexibility and can lead to improved generative models. The paper presents experimental results on CIFAR-10 to support this claim. However, the evidence might be considered somewhat limited due to the restriction to a single, relatively low-resolution dataset. The lack of thorough ablation studies on the design choices of the coefficient parameterization also weakens the evidence for the specific design being optimal.

**Essential References Not Discussed:**

The paper include the related references.

**Experimental Designs Or Analyses:**

The experimental design and analyses raise several concerns:

- Limited Dataset: The evaluation is limited to CIFAR-10 (32x32 images), which is insufficient to demonstrate the effectiveness of the proposed method on more complex and higher-resolution datasets. I suspect that the increased dimensionality of the 2D time sampling might lead to optimization difficulties as the image size grows.
- Lack of Ablation: The absence of ablation studies on the design choices for the coefficient parameterization is a significant oversight. It's quite hand-crafted in terms of the specific parameterization in equation 8, and whether other designs might yield better results.
- Inconsistent Results: There are discrepancies in the reported performance across different tables (e.g., Table 2, 3, and 4 compared to Table 5), and it's unclear if different models or settings were used. The comparison in Tables 2-4 is limited to NFE=10, which is unusually low for standard diffusion and flow models to achieve good performance.
- Missing Baselines: Table 5 lacks many baseline results, making it difficult to properly assess the improvement offered by the proposed method. Furthermore, the comparison only includes distillation methods and inexplicably omits 1-Rectified Flow (Flow Matching), which seems like a relevant baseline.

**Methods And Evaluation Criteria:**

The proposed method, involving the joint training of a diffusion/flow model, a discriminator, and a coefficient model with a 2D time sampling, presents an interesting alternative to standard interpolation techniques. The evaluation seems to primarily rely on FID scores obtained on the CIFAR-10 dataset. While FID is a common metric for generative model evaluation, the limited scope of the experiments raises questions about the generalizability of the findings.

**Other Comments Or Suggestions:**

Please see the questions part.

**Other Strengths And Weaknesses:**

Strengths: The paper proposes a novel idea of using a learned 2D interpolation for diffusion and flow models, which could potentially offer more flexibility than standard interpolation.

Weaknesses: Please see the questions below

**Questions For Authors:**

1. In the introduction, you mention a connection to Neural ODEs. Could you please elaborate on the nature of this relationship and how your work builds upon or relates to concepts from Neural ODEs?
2. Why was the specific parameterization for the coefficient model chosen as described in equation 8? Were other parameterizations explored, and if so, what were the results? An ablation study on this design choice would be beneficial.
3. The experimental evaluation is currently limited to CIFAR-10. Do you have plans to extend the evaluation to higher-resolution datasets like Imagenet 64x64 or include more comprehensive comparisons on datasets like FFHQ or AFHQ? Given the 2D time sampling, are there potential scalability challenges with increasing image size?
4. In equation 5, x_T is used but not defined. Similarly, in Algorithm 1, ρ_T is used without definition. What are these variables, and what is the difference between ρ_1 and ρ_T?
5. From Section 3.2.2 it seems that the method is currently applied to diffusion models predicting noise. Could you clarify if this approach can be extended to other types of diffusion and flow models, such as those predicting velocities?
6. Tables 2, 3, and 4 present results with NFE=10, which is unusually low for standard DDPM and Flow Matching methods. The performance also differs from Table 5. Were different models or training settings used in these tables? Please clarify these discrepancies.
7. Table 5 is missing many baseline results, including 1-Rectified Flow (Flow Matching). Could you provide these missing results for a more comprehensive comparison?
8. What is the primary motivation for using adversarial training in your method? Could you compare the performance of your method with and without the adversarial loss to isolate the impact of the proposed interpolation technique?  It's unclear whether the performance gains are solely due to the proposed interpolation method or if the adversarial training plays a significant role, especially in terms of FID improvement.
9. In Table 6, the FID scores for the first two rows (presumably vanilla EDM) are significantly higher than the EDM results in Table 5. Could you explain this difference? I assume the first row is vanilla EDM so it should match table 5 row 3 in diffusion models?

**Relation To Broader Scientific Literature:**

Previous work like rectified flow (flow matching), DDPM or other variants mainly focus on using a predefined and fixed intepolation. This work extend it using a learned intepolation and expand it to 2D

**Theoretical Claims:**

This paper doesn't include new proofs or theoretical claims

---

### Decision · Program_Chairs · 2025-05-01

**Decision:**

Accept (poster)

**Comment:**

Standard diffusion and flow-based models interpolate between noise and data, and the interpolating coefficients are usually shared across dimensions. This paper proposes a simple method for having dimension-dependent interpolation coefficients.

The paper received three reviews, yet one was flagged as being AI-generated. Upon inspecting the review, I believe the flag to be very plausible since the review explicitly mentioned the authors conducting experiments only on CIFAR-10 despite this not being the case. I thus ignored this review in my decision making process, and instead read the paper and engaged with the authors.

Reviewers found the method novel and sensible, but criticized the lack of experiments on ImageNet. The authors provided these experiments in their rebuttal in a way that I believe satisfactorily addressed the main criticisms raised. Overall I believe this paper should be accepted: it presents a simple yet well-motivated and novel method which achieves good empirical results.